# Translating new evidence into clinical practice: a quasi-experimental controlled before–after study evaluating the effect of a novel outreach mentoring approach on knowledge, attitudes and confidence of health workers providing HIV and infant feeding counselling in South Africa

Ameena Goga [1,2,3] Tanya Doherty, [1,4,5] Samuel Manda, [6,7] Tshifhiwa Nkwenika, [6] Lyn Haskins, [8] Vaughn John, [9] Ingunn M S Engebretsen [10] Ute Feucht, [2,11,12,13] Ali Dhansay, [14,15] Nigel Rollins, [16] Max Kroon, [17,18] David Sanders, [4,19] Shuaib Kauchali, [20] Thorkild Tylleskär, [10] Christiane Horwood [8]

David Sanders

For numbered affiliations see end of article.

**Correspondence to**
Dr Ameena Goga;
Ameena.Goga@mrc.ac.za

## ABSTRACT

**Objectives** We report the effectiveness of a mentoring approach to improve health workers' (HWs') knowledge, attitudes and confidence with counselling on HIV and infant feeding.

**Design** Quasi-experimental controlled before–after study.

**Setting** Randomly selected primary healthcare clinics (n=24 intervention, n=12 comparison); two districts, South Africa.

**Participants** All HWs providing infant feeding counselling in selected facilities were invited.

**Interventions** Three 1–2 hours, on-site workshops over 3–6 weeks.

**Primary outcome measures** Knowledge (22 binary questions), attitude (21 questions—5-point Likert Scale) and confidence (19 questions—3-point Likert Scale). Individual item responses were added within each of the attitude and confidence domains. The respective sums were taken to be the domain composite index and used as a dependent variable to evaluate intervention effect. Linear regression models were used to estimate the mean score difference between intervention and comparison groups postintervention, adjusting for the mean score difference between them at baseline. Analyses were adjusted for participant baseline characteristics and clustering at health facility level.

**Results** In intervention and comparison sites, respectively: 289 and 131 baseline and 253 and 114 follow-up interviews were conducted (August–December 2017). At baseline there was no difference in mean number of correctly answered knowledge questions; this differed significantly at follow-up (15.2 in comparison; 17.2 in intervention sites (p<0.001)). At follow-up, the mean attitude and confidence scores towards breast feeding were better in intervention versus comparison sites (p<0.001 and p=0.05, respectively). Controlling for confounders, interactions between time and intervention group and preintervention values, the attitude score was

### Strengths and limitations of this study

► Fieldwork was conducted in two geographically and historically different provinces, facilitating generalisability of results.

► The intervention was participatory, low-intensity, on-site and integrated into routine services.

► Several types of analyses were conducted which all yielded congruent results.

► However, limitations were that (1) We purposively selected districts for inclusion, (2) We could not control for health workers' (HWs') personal breastfeeding experience as we did not gather these data, (3) The follow-up evaluation was undertaken 3 months after the intervention—thus, we measured short-term benefits, (4) We did not measure the direct effect of improved HWs' knowledge, attitudes and confidence on HWs' counselling and mothers' infant feeding practices and (5) We did not co-design the intervention with women living with HIV. Co-designing the intervention with women living with HIV may have resulted in a different intervention and results, and needs to be undertaken in future work.

► The finding that knowledge scores among participants who attended three workshops were significantly better than knowledge scores among participants who attended less than three workshops, may simply reflect better motivation among attendees of more workshops, rather than the effect of the workshops themselves. We could not tease out these effects.

5.1 points significantly higher in intervention versus comparison groups.

**Conclusion** A participatory, low-intensity on-site mentoring approach to disseminating updated infant feeding guidelines improved HWs' knowledge, attitudes

and confidence more than standard dissemination via a circular. Further research is required to evaluate the effectiveness, feasibility and sustainability of this approach at scale.

## INTRODUCTION

The benefits of breast feeding in all settings, and particularly in low-income middle-income settings with high HIV prevalence, are undisputed.[1 2] Policies and clinical practice guidelines on preventing vertical transmission (PVT) of HIV, also known as preventing mother to child transmission (PMTCT) of HIV and infant feeding, have undergone frequent evidence-based revisions. For example South African PMTCT policy and its accompanying infant feeding recommendations have been revised five times since 2001 (2008, 2010, 2013, 2015, 2019).[3–7] Additionally, in 2011 a national infant feeding declaration withdrew free commercial infant formula as part of the PVT programme,[8] and in 2017 the infant and young child feeding policy was updated to recommend that women living with HIV may continue breast feeding for up to 24 months or longer (similar to the general population) while being fully supported for antiretroviral therapy (ART) adherence. This followed a 2016 WHO update which also stated that mixed feeding is not a reason to stop breast feeding in the presence of antiretroviral (ARV) drugs.[9] However, a key gap is that these policies have not been effectively communicated to all health workers (HWs)—a requirement of the mother-baby friendly initiative.[10 11] HWs play a critical role in guiding infant feeding choices and sustaining infant feeding practices;[11–13] they wield power and authority[12 14] but their potentially positive influence on infant feeding is compromised by confusion over HIV and infant feeding, which has eroded their own confidence.[11 13] Identifying and implementing optimal strategies to effectively disseminate updated guidelines have lagged behind. Multicomponent dissemination strategies, which aim to increase the reach, ability and motivation of HWs, are more effective than one strategy alone.[15] However, in reality there are few published studies that inform guideline dissemination. Most of these are from high-income settings and may not be relevant to low-income settings which have unique challenges.[15]

Research has demonstrated that improving HWs' capacity can significantly improve their skills, self-efficacy and confidence to counsel, support and promote breast feeding among women living with HIV.[16 17] Consequently, a key question was: What learning approach could best develop HW capacity and confidence to implement the updated HIV and infant feeding guideline, using a methodology that is sustainable and feasible to implement at scale? Pedagogical research highlights the advantage of participatory training compared with standard didactic teaching for improving HW skills.[18 19] Thus, we sought to determine whether a participatory outreach mentorship approach to disseminate the updated HIV and infant feeding guidelines, using simple low-technology activities, improves HWs' knowledge of, attitudes towards and

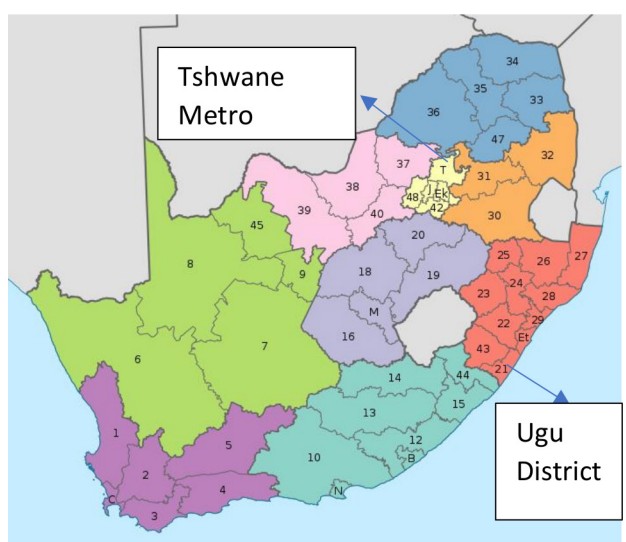

**Figure 1** Study districts: Tshwane Metropolitan Municipality in Gauteng Province and Ugu District in KwaZulu-Natal Province of South Africa.

confidence with counselling on HIV and infant feeding. We chose to focus on HWs' knowledge, attitudes and confidence as HWs in South Africa consider themselves as advocates for babies.[20] Additionally, they are one of the key influential groups in the complex socioecology of infant feeding.[12 13 21 22]

## METHODS

### Study design

A quasi-experimental before–after design with intervention and comparison sites was used. Two purposively selected districts (Ugu District and Tshwane Metro/ District) in South Africa in each of two geographically disparate provinces, KwaZulu-Natal (KZN) and Gauteng (figure 1), were included. Both provinces experienced a policy change in June 2017, when the 2013 South African Infant and Young Child Feeding Policy was amended to align with the 2016 WHO/UNICEF update on HIV and infant feeding .

### Sampling

In Ugu District all four subdistricts were selected; within Tshwane District two of the seven service delivery regions were randomly selected.

Twelve intervention and six comparison primary healthcare (PHC) clinics were randomly sampled in Ugu District and Tshwane District (separately). Only clinics with above the median number of annual clinic visits for children under 5 years in the district were eligible for inclusion in the sampling frame. The comparison clinics served to capture any temporal changes in HW knowledge, confidence and attitudes due to other interventions or trainings; hence a smaller sample was required in comparison versus intervention sites as the latter required more precise estimates of the intervention effect. A two-stage process was used to recruit participants. First, research

staff explained the study and participant inclusion and exclusion criteria to each facility manager during face-to-face on-site introductory meetings. The facility manager compiled a list of all eligible HWs involved in the care of pregnant women and children, including nurses, midwives, visiting doctors, lay counsellors, dieticians, nutritionists, facility managers and community health workers (CHWs). In the second stage, research staff approached eligible HWs and invited them to participate in the research. We aimed to recruit a manageable size of 8–10 HWs per clinic for participation in the intervention, and in the evaluation. The same staff were approached for the baseline and follow-up evaluations.

## Sample size

The sample size was determined based on 80% power and α 0.05 to measure a 15 percentage points difference in HW confidence in HIV and infant feeding counselling between the intervention and comparison clinics comparing baseline and follow-up. The expected effect was based on the researchers' experience and data from recent studies in South Africa with the baseline level of high confidence to counsel HIV-positive women on breastfeeding duration set at 45%.[23] It was assumed that the confidence score would remain unchanged in the comparison clinics, implying a two-sample test in the postintervention period. Clinic-level analyses were used for the sample size calculations, assuming a sampling ratio of 2:1 for the intervention clinics and a SD of 15% in the mean score between clinics. Based on these assumptions, and adjusting for clustering, the sample size was determined to be 24 intervention clinics and 12 comparison clinics.[24] Within the intervention and comparison clinics, all HWs (nurses, midwives, visiting doctors, lay counsellors, dieticians, nutritionists, facility managers and CHWs), involved in caring for pregnant women and children were invited to participate in the study—we anticipated a mean number of HWs per participating facility to be 8–10.

## Description of the intervention

We designed a participatory intervention comprising on-site mentoring through three workshops in each clinic,

involving 303 selected HWs who provide care for pregnant women, breastfeeding mothers and their infants. This mentoring approach had five distinct features: (1) On-site: learning occurred in context; (2) Open to all cadres of HWs; (3) Team-based, participants learnt together; (4) Content was led by self-identified gaps in knowledge; and (5) Activities were piloted and rooted in a theoretical framework. The intervention was delivered by the same trained facilitator (a nurse in Gauteng and nutritionist in KZN) in each intervention clinic. Each workshop lasted 1–2 hours and three were conducted over a period of 3–6 weeks; all had well-defined learning outcomes. The intervention has been described elsewhere.[25] In summary, our participatory intervention was guided by evidence that HWs' attitudes and practices are influenced by various factors, not just exposure to training and information.[26] We used Dee Fink's six-part taxonomy as a guiding theory. This proposes that significant learning only occurs by developing foundational knowledge, applying skills, integrating ideas, developing new feelings/interests and values, and learning how to learn (encouraging the spirit of enquiry) (figure 2).[27] Additionally, we applied the Theory of Planned Behaviour to the intervention design (figure 2).[28 29] This states that an individual's intention to perform a behaviour is influenced by the person's attitudes towards performing the behaviour, their beliefs about whether people who are important to them will approve of the behaviour (subjective norms) and their beliefs about how likely they are to implement the behaviour successfully. According to this theory, if HWs are to provide infant feeding counselling and support in accordance with updated infant feeding guidelines to HIV-positive or HIV-negative mothers, they need to agree with the change, believe that their colleagues and other stakeholders will approve of the action, and believe in their ability to implement it successfully. The workshops were tailored to achieve these goals: workshop 1 covered knowledge gaps reported by participants, controversial statements and advantages of breastfeeding. Following workshop 1, a poster or cards with key messages were placed in a prominent place in the clinic. Workshop 2 comprised a progressive case study discussed

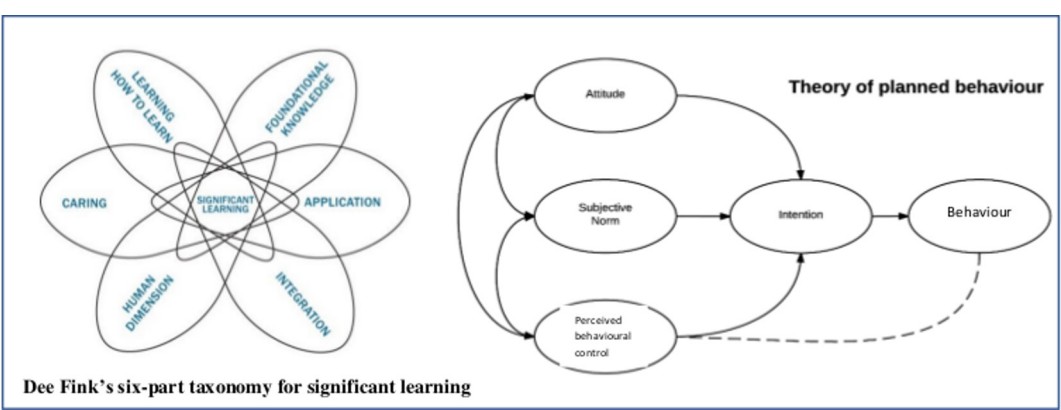

**Figure 2** Theoretical frameworks which informed the development of the intervention.

in small groups. Workshop 3 involved one-to-one mentorship: each participant was observed providing infant feeding counselling or a case study was discussed if no mothers were available for counselling. The same facilitator conducted all three workshops at each clinic. In addition, a WhatsApp cellphone messaging group was established to support participants in intervention sites to facilitate sharing of concerns, tips for counselling and dealing with difficult situations. Key messages were posted on the group approximately weekly. Comparison and intervention subdistricts were exposed to routine supervision and training activities that took place at district level. The study team documented that the June 2017 circular issued by the National Department of Health, informing health facilities of the change in the infant and young child feeding policy, was disseminated to comparison clinics as an announcement via email and other electronic communication as well as during meetings or trainings. We documented that in Tshwane, 15 of the 18 clinics had received the circular; 11 via email and 3 at a meeting. In Ugu 9 of 17 clinics had received the circular; 8 received it via hand delivery and 1 via email.

### Patient and public involvement

Patients and the public were not involved in the design of this study, as the main population of interest were HWs. The intervention and tool were piloted among a separate group of HWs to determine length, complexity of questions and level of understanding. These details are explained in our intervention paper.[25]

### Data collection

Data were collected between August and December 2017 by dedicated trained non-nurse data collectors who were independent of the intervention staff. As per study design, data collection staff were not part of any intervention activities and had never been exposed to the intervention. The primary outcome measure for the study was the confidence level of HWs to counsel on infant feeding, evaluated using a Likert Scale tool, developed after reviewing the WHO breastfeeding counselling course, and the WHO HIV and infant feeding counselling course[9 30–33] (see tool in online supplemental material). Secondary outcomes included HW knowledge and attitude about breastfeeding counselling. A baseline assessment among all participating HWs in intervention and comparison sites was undertaken prior to the start of the intervention (August 2017). HWs self-completed the assessment on study-provided electronic tablets at their workplaces. Questions covered basic demographic information, types of activities undertaken at work, knowledge, attitudes and confidence around counselling on infant feeding (see tool in online supplemental material). Approximately 12 weeks after the baseline assessment, a follow-up assessment using the same tool was conducted among the same group of HWs. HWs who were not in the clinic on the day of the follow-up assessments were included in a special catch-up assessment. Questionnaire

software had built in range and skip logic and data were transferred automatically to a database held at the University of KwaZulu-Natal.

### Data analysis

There were three outcomes in the study: (a) 22 knowledge statements which were scored 1 if correctly answered and 0 if not; evaluation of answers were based on existing literature and guidelines (binary outcomes); (b) 21 attitude questions whose responses were measured on a 5-point Likert Scale—given as completely disagree (1); disagree (2); neutral (3); agree (4) and completely agree (5); positive attitudes received higher scores; and (c) 19 statements on confidence item questions which were also measured on a Likert Scale, scored as such: not at all confident (1), not confident (2), confident (3) and very confident (4). For both attitude and confidence domains, a participant outcome was measured by the sum of the responses to the respective items (we verified that there was not a missing response on the items). Thus, the ranges for the attitude and confidence scores were 21–105, and 19–75, respectively.

Participants' baseline and follow-up characteristics and outcomes between the intervention and control areas were compared using $\chi^2$ tests for categorical variables and two-sample t-tests for continuous measures, after confirming that data were normally distributed. To assess the effect of the proposed intervention, several analysis methods for comparing intervention effect in before (pre)–after (post) quasi-experimental designs were considered. These included using postmeasures and change from preintervention to postintervention as the response variables. These approaches that use change and postmeasurements as the outcome, adjusting for preintervention measurements are recommended, and often give similar results.[34] In this paper, we considered three methods for estimating and testing the intervention effect using the sum of individual attitude or confidence scores as an outcome variable in a linear regression. The first method used the postintervention measurements as the outcome variable but adjusted for the preintervention values; the second method analysed the change score as an outcome variable adjusting for pretreatment values. The third method analysed the vectors of premeasurements and postmeasurements as the outcome variable, and used time (coded 1 at follow-up and 0 at baseline) and treatment (coded 1: intervention group and 0: comparison group) as a covariates with an interaction term for time and treatment, in addition to an adjustment for pretreatment values). Using methods 1 and 2 the coefficient for the intervention estimated the differences in the postintervention means and differences in the mean change of sum scores mean between the groups, controlling for the preintervention measurement. Using the third method, the sum of coefficients of intervention and the interaction terms was taken as the mean difference between groups post-treatment, allowing for pretreatment mean differences between the groups. All analyses also controlled

INTERVENTION SITES

COMPARISON SITES

Baseline

Baseline

Baseline n= 420 interviews

23 clinics; n= 289 interviews
(Ugu District n= 137 interviews
Tshwane District n= 152 interviews)

12 clinics; n= 131 interviews
(Ugu District n= 75 interviews
Tshwane District n= 56 interviews)

36 (12.5%) LTFU

17 (13.0%) LTFU

Follow-up
23 clinics; n= 253 interviews (87.5%)
(Ugu District n= 120 interviews
Tshwane District n= 133 interviews)

Follow-up
12 clinics; n= 114 interviews (87.0%)
(Ugu District n= 64 interviews
Tshwane District n= 50 interviews)

Follow-up n= 367 interviews

LTFU: lost to follow up

**Figure 3** Study population at baseline and follow-up for intervention and comparison sites.

for baseline participant characteristics and prior training. Analyses adjusted for possible clustering effect at the site level, using a variance-correction method.[24] All the treatment effect comparison analyses were done on an intention-to-treat, rather than per-protocol, basis. Data can be obtained by emailing the corresponding author.

### Ethics
Permission for undertaking the study was obtained from Tshwane and Ugu Districts. Informed consent was sought from all study participants and no personal identifying information was captured in the questionnaires, only a study identification number.

### RESULTS
At baseline and follow-up, 23 intervention clinics (one large clinic was sampled twice with two rounds of data collection per time point) and 12 comparison clinics were visited; 289 and 131 HW interviews were conducted at baseline in intervention and comparison clinics, respectively (figure 3). Loss to follow-up between baseline and follow-up did not differ between intervention and comparison sites (17 (13.0%) in comparison sites vs 36 (12.5%) in intervention sites).

Tshwane and Ugu Districts did not differ in the main outcome measures at baseline (knowledge, attitude and confidence). Additionally, they were similar in all HW characteristics except three: Tshwane had significantly more participants with less than 2 years employment (14.4% vs 6.2%, p=0.007), more registered nurses (57% vs 26.2%, p=0.02) and fewer lay counsellors/CHWs (7.3% vs 50.0%, respectively, p=0.02). Given the lack of a statistically significant difference in the main outcome variables at baseline, data from the two sites were combined for the analysis.

All staff approached agreed to participate in the interviews. There were no statistically significant differences between intervention and comparison sites at baseline, regarding district of origin, median age of respondent, gender, cadre of HW and working duration (table 1). The proportion of participants who had received previous training (through the routine health system) on specific topics was similar in intervention versus comparison sites, except for three topics which had better coverage in comparison sites (online supplemental figure 1). These were: ever trained on how to assess and support ART adherence for HIV-positive women (78.6% in intervention sites vs 89.2% in comparison sites, p=0.01); ever trained about managing breastfeeding problems (76.5% in intervention sites and 86.2% in comparison sites, p=0.02); and received any information or training about the revised infant feeding policy (55.1% in intervention sites vs 67.4% in comparison sites, p=0.02). At baseline, activities around breastfeeding counselling and management were similar between comparison and intervention sites in all respects except that comparison site participants reportedly spoke more frequently to HIV-positive pregnant women about feeding than intervention participants (67% vs 71.6% spoke more than 1–3 times per month, p=0.04, data not shown).

In intervention sites, workshops were attended by 84%–88% of participants interviewed at follow-up (table 2).

### Effect of the intervention on HW knowledge
At baseline, knowledge about key infant feeding statements or facts was similar between intervention and comparison sites, except for knowledge about soft porridge (table 3). Although at baseline, more than 90% of intervention and comparison site participants knew that a baby under 4 months should not be given soft porridge if hungry, significantly more intervention site participants knew this recommendation (table 3). The percentage of participants at baseline correctly answering the more difficult

**Table 1** Characteristics of the participants in the intervention and comparison groups at baseline

| Characteristic | Intervention group (n=289) (N (%)) | Comparison group (n=131) (N (%)) | P value |
|---|---|---|---|
| District | | | 0.06 |
| Tshwane | 152 (52.6) | 56 (42.8) | |
| Ugu | 137 (47.4) | 75 (57.3) | |
| Age categories | | | 0.11 |
| 23–35 years | 56 (19.4) | 38 (29.7) | |
| 36–41 years | 61 (21.2) | 25 (19.5) | |
| 42–46 years | 53 (18.4) | 26 (20.3) | |
| 47–54 years | 64 (22.2) | 18 (14.1) | |
| Over 54 years | 54 (18.8) | 21 (16.4) | |
| Gender | | | 0.66 |
| Female | 267 (92.7) | 118 (91.5) | |
| Male | 21 (7.3) | 11 (8.5) | |
| Cadre of health worker | | | 0.05 |
| Community-level worker | 84 (29.5) | 52 (40,0) | |
| Trained health professional* | 151 (53.0) | 64 (49.2) | |
| Enrolled nurse | 50 (17.4) | 14 (10.8) | |
| Work experience in year/years | | | 0.20 |
| Less than 1 year | 4 (1.4) | 3 (2.3) | |
| 1 to <2 years | 23 (8.0) | 12 (9.3) | |
| 2 to < 5 years | 36 (12.5) | 18 (14.0) | |
| 5 to <10 years | 71 (24.7) | 43 (33.3) | |
| 10 years or more | 154 (53.5) | 53 (41.1) | |

*Includes 68% nurses in the intervention arm and 58% nurses in the comparison arm. This group also includes operation managers, dieticians, doctors and nutritionists.

questions (on bottle sterilisation, storing expressed breast milk, feeding HIV-exposed infants) was low (table 3). At follow-up significantly more intervention site participants correctly answered knowledge questions, regarding the leading cause of death in children under 5 years, the risk of formula feeding, duration of breast feeding for

HIV-negative mothers and women living with HIV, how to stop breast feeding, complementary feeding, storing expressed breast milk, feeding while at work, breast feeding and viral suppression, mixed feeding in women living with HIV, adherence to ART and breast feeding, breastfeeding difficulties in women living with HIV and managing women living with HIV who are breast feeding, than comparison site participants (table 3). The significant differences between intervention and comparison sites regarding soft porridge were not present at follow-up. Although improvements were seen in knowledge relating to the risks of mixed feeding for women living with HIV, most HWs still provided incorrect responses at follow-up. At baseline, the mean number of correctly answered knowledge questions was 15.0 (68%) in comparison sites versus 15.2 (69%) in intervention sites, p=0.89 (table 3). At follow-up the mean number was 15.2 (69%) in comparison sites and 17.2 (78.2%) in intervention sites, p<0.001 (table 3). For two questions measuring knowledge about the 2017 change in infant feeding guidelines, namely, 'Continued breastfeeding for 2 years is the recommended infant method in SA for ALL children, regardless of mother's HIV status' and 'In South Africa, HIV-infected women who are breastfeeding should be supported to adhere to antiretroviral treatment and should introduce complementary foods at 6 months and be supported to continue breastfeeding for at least 2 years. (True)', there was a 36% improvement in knowledge in the intervention group at follow-up compared with a 13% increase in knowledge in the control group. For the second question there was a 15% increase in correct knowledge in the intervention group at follow-up while for the comparison group knowledge decreased from 89% to 81%. At follow-up, knowledge scores of participants who attended three workshops compared with knowledge scores of participants who attended less than three workshops was significantly better (p<0.001).

### Effect of the intervention on attitudes

At baseline, intervention and comparison sites were similar in HW attitudes except for attitudes towards feeding a crying baby and expressing breast milk, which

**Table 2** Attendance at workshops 1–3 measured at follow-up in intervention sites

| Number of staff attending each workshop | Attended workshop, n | Attended catch-up, n | Total attended n/N (%) | |
|---|---|---|---|---|
| Group workshop 1 | 202 | 63 | 265/303 (87.5) | |
| Group workshop 2 | 223 | 34 | 257/303 (84.8) | |
| Workshop 3 (clinical mentoring) | 216 | 40 | 256/303 (84.5) | |
| **Number of workshops attended** | | | **Number** | **%** |
| No workshop | | | 42 | 13.9 |
| One workshop | | | 8 | 2.6 |
| Two workshops | | | 6 | 2.0 |
| All three workshops | | | 247 | 81.5 |
| **Total** | | | **303** | **100** |

**Table 3** Knowledge of health workers about breast feeding in the intervention and comparison sites at baseline and follow-up

| Knowledge statements | Number (%) with correct answers at baseline | | | Number (%) with correct answers at follow-up | | |
|---|---|---|---|---|---|---|
| | Intervention (n=289) | Comparison (n=131) | P value* | Intervention (n=250) | Comparison (n=112) | P value* |
| **Knowledge relating to updates in the HIV and infant feeding guidelines** | | | | | | |
| *Significant improvements between intervention and comparison groups at follow-up* | | | | | | |
| Continued breast feeding for 2 years is the recommended infant method in SA for *all* children, regardless of mother's HIV status (true)† | 190 (65.7) | 91 (70.0) | 0.39 | 224 (89.6) | 88 (78.6) | <0.01 |
| An HIV-positive mother who is virally suppressed on antiretroviral treatment should breast feed her child rather than not breast feed to improve the child's survival (true)† | 237 (82.0) | 108 (83.1) | 0.79 | 236 (94.4) | 96 (85.7) | <0.01 |
| A mother who has missed six tablets of fixed-dose combination antiretroviral therapy in 1 month is considered to be poorly adherent and should stop breast feeding immediately (false)†‡ | 181 (62.6) | 89 (68.5) | 0.25 | 201 (80.4) | 72 (64.3) | <0.01 |
| In South Africa, HIV-infected women who are breast feeding should be supported to adhere to antiretroviral treatment and should introduce complementary foods at 6 months and be supported to continue breast feeding for at least 2 years (true)† | 245 (84.8) | 116 (89.2) | 0.22 | 244 (97.6) | 91 (81.3) | <0.01 |
| When an HIV-infected mother is ready to add complementary feeds she should stop breast feeding rapidly over a 24-hour period (false)† | 214 (74.1) | 103 (79.2) | 0.25 | 217 (86.8) | 86 (76.8) | <0.05 |
| If a mother misses two doses of her antiretroviral therapy in 1 month, she should be classified as a treatment failure (false)†‡ | 185 (64.0) | 82 (63.1) | 0.85 | 191 (76.4) | 73 (65.2) | <0.05 |
| *Low levels of knowledge (<80%) at baseline in both groups—no significant differences between intervention and comparison groups at follow-up (concept that this relates to)* | | | | | | |
| If an HIV-exposed baby is receiving both breast milk and formula milk, the mother should choose either breast feeding or formula feeding if she is adherent to antiretroviral therapy (false)† (mixed feeding with formula and breast milk) | 69 (23.9) | 29 (22.3) | 0.71 | 75 (30.0) | 28 (25.0) | 0.33 |
| A mother living with HIV and adherent to antiretroviral treatment cannot exclusively breast feed her 4-month-old infant because she is working. It is better for this mother to give formula during the day and breast feed at night rather than giving no breast milk at all (true)† (mixed feeding with formula milk and breast milk) | 22 (7.6) | 14 (10.8) | 0.29 | 40 (16.0) | 14 (12.5) | 0.38 |
| *High levels of knowledge (≥80%) at baseline in both groups—no significant differences between intervention and comparison groups at follow-up* | | | | | | |
| In South Africa, HIV-infected women who are breast feeding should be supported to adhere to antiretroviral treatment and should be counselled and supported to exclusively breast feed their infants for the first 6 months of life while maintaining an undetectable viral load (true)† | 281 (97.2) | 123 (94.6) | 0.18 | 242 (96.8) | 109 (97.3) | 0.79 |
| Mothers living with HIV who are receiving antiretroviral treatment and are virally suppressed should be advised not to breast feed their infants (false)† | 252 (87.2) | 115 (88.5) | 0.85 | 230 (92.0) | 99 (88.4) | 0.27 |
| **General breast feeding** | | | | | | |
| *Significant improvements between intervention and comparison groups at follow-up* | | | | | | |
| In South Africa, the leading cause of death among children under 5 years is pneumonia (true) | 189 (65.4) | 82 (63.1) | 0.65 | 230 (92.0) | 75 (67.0) | <0.01 |
| Giving any formula milk during the first 6 months of life increases the risk of death from diarrhoea and/or pneumonia (true) | 246 (85.1) | 104 (80.0) | 0.17 | 232 (92.8) | 95 (84.8) | 0.02 |

Continued

**Table 3** Continued

| Knowledge statements | Number (%) with correct answers at *baseline* | | | Number (%) with correct answers at *follow-up* | | |
|---|---|---|---|---|---|---|
| | Intervention (n=289) | Comparison (n=131) | P value* | Intervention (n=250) | Comparison (n=112) | P value* |
| It is safe to give the baby expressed breast milk that has been kept outside the fridge for 8 hours (true) | 106 (36.7) | 43 (33.1) | 0.48 | 120 (48.0) | 38 (33.9) | <0.05 |
| A mother who is working and giving formula milk should mix the milk herself and leave for the carer to give during the day (false)‡ | 218 (75.4) | 94 (72.3) | 0.50 | 189 (75.6) | 68 (60.7) | <0.01 |
| *Low levels of knowledge (<80%) at baseline in both groups—no significant differences between intervention and comparison groups at follow-up* | | | | | | |
| When sterilising feeding bottles cover the bottles with water in a saucepan and place on the heat. As soon as the water boils remove from heat and do not leave the bottles in the water until completely cool (false)‡ | 64 (22.2) | 27 (20.8) | 0.75 | 53 (21.2) | 25 (22.3) | 0.81 |
| *High levels of knowledge (≥80%) at baseline in both groups—no significant differences between intervention and comparison groups at follow-up* | | | | | | |
| Exclusive breast feeding is the recommended infant feeding method for *all* infants aged 0–6 months in SA, regardless of mother's HIV status (true) | 271 (93.8) | 118 (90.8) | 0.27 | 234 (93.6) | 102 (91.1) | 0.32 |
| A baby under 4 months should be given soft porridge once he/she seems hungry (false)‡ | 284 (98.3) | 124 (95.4) | 0.09 | 247 (98.8) | 108 (96.4) | 0.13 |
| Giving a baby expressed breast milk is not as good as breast feeding (false)‡ | 234 (81.0) | 106 (81.5) | 0.89 | 218 (87.2) | 96 (85.7) | 0.70 |
| There are long-term health benefits of breast feeding for mother and child that last beyond the breastfeeding period (true) | 264 (91.4) | 116 (89.2) | 0.49 | 232 (92.8) | 100 (89.3) | 0.26 |
| **Breast feeding and HIV** | | | | | | |
| *Significant improvements between intervention and comparison groups at follow-up* | | | | | | |
| An HIV-positive mother who has cracked nipples should continue to breast feed unless they are bleeding (true) | 143 (49.5) | 64 (49.2) | 0.96 | 187 (74.8) | 59 (52.7) | <0.01 |
| If a baby has a positive PCR (HIV test) at birth the mother should stop breast feeding if this is affordable and feasible in her situation (false)‡ | 224 (81.0) | 100 (76.9) | 0.90 | 214 (85.6) | 82 (73.2) | <0.01 |
| *High levels of knowledge (<80%) at baseline in both groups—no significant differences between intervention and comparison groups at follow-up* | | | | | | |
| An HIV-exposed baby who is exclusively breast feeding should be given some water when the weather is very hot (false)‡ | 270 (93.4) | 122 (93.9) | 0.87 | 239 (95.6) | 105 (93.8) | 0.45 |
| **Mean Knowledge Score (SD) out of 22** | 15.2 (2.6) | 15.0 (3.1) | 0.61* | 17.2 (2.1) | 15.2 (2.8) | <0.01 |

The tables displays numbers with correct knowledge.

*Independent t-test comparing intervention and comparison sites at the relevant time point.

†These questions measure the change in knowledge relating to the South African Department of Health June 2017 circular and the WHO 2016 updated HIV and infant feeding guidelines.

‡The statement is false; thus, the scales were inverted during data analysis.

were significantly better in intervention sites (online supplemental table 1). At follow-up, attitudes to breast feeding and HIV were significantly better in the intervention group for 13 of the 21 questions and the mean attitude score towards breast feeding was significantly better in intervention sites (p<0.001) (online supplemental table 1). At follow-up, HWs in the intervention group were significantly less confused about what to tell women living with HIV about infant feeding . Methods 1 and 2 yielded the same results, except for the effect of baseline attitude score. Thus, in table 4 below, we only show results for Methods 1 and 3. Controlling for other variables, postintervention attitude was significantly better in intervention, compared with comparison, sites (table 4). Using Method 1, attitude at follow-up was 5.4 points higher in the intervention group than in the comparison group; Method 3 analysis showed a significant 5.1-point higher score in the intervention compared with the comparison group. Using Method 1, being an enrolled nurse, and being in the youngest (36–41 years) or oldest (>54 years) age group was associated with a significantly lower attitude

**Table 4** Adjusted effect of the intervention on health worker attitude score using different methods (effect estimate and 95% CI)

| Variable | Method 1 | | Method 3 | |
|---|---|---|---|---|
| | Effect estimate | 95% CI | Effect estimate | 95% CI |
| Attitude score at baseline | 0.5 | 0.3 to 0.7* | N/A | N/A |
| Intervention | 5.4 | 3.9 to 6.9* | 5.1 | 2.1 to 8.1* |
| Follow-up period | N/A | N/A | 1.8 | 0.2 to 3.4* |
| Professional role versus community level | | | | |
| Trained health professional | 1.6 | −0.05 to 3.2 | 4.8 | 2.8 to 6.7* |
| Enrolled nurse | −2.4 | −5.0 to −0.2* | 0.9 | −1.4 to 3.2 |
| Ugu District versus Tshwane District | −0.83 | −2.2 to 0.5 | −1.4 | −3.1 to 0.2 |
| Age category versus 23–35 years | | | | |
| 36–41 years | −2.8 | −5.4 to −0.2* | −1.8 | −4.1 to 0.6 |
| 42–46 years | −0.9 | −3.3 to 1.5 | −0.2 | −2.5 to 2.2 |
| 47–54 years | 0.5 | −2.0 to 2.9 | −1.2 | −3.1 to 0.8 |
| Over 54 years | −3.3 | −5.7 to −1.0* | −2.2 | −4.8 to 0.3 |
| Work experience <5 years vs ≥5 years | −0.3 | −2.5 to 1.9 | −1.3 | −3.4 to 0.8 |
| Received training or information at work about the revised policy | 0.5 | −1.4 to 2.3 | 1.7 | 0.1 to 3.24 |
| Received any training about managing common breastfeeding problems? | 0.3 | −2.5 to 3.0 | 3.2 | 0.9 to 5.5 |
| Ever received any training about how to assess and support antiretroviral therapy adherence for HIV-positive women? | −0.1 | −2.2 to 2.0 | 1.6 | −0.5 to 3.8 |

All analyses are adjusted for clustering.
*p<0.05.
N/A, not applicable.

score. Using Method 3, trained health professionals had a significantly higher attitude score at follow-up (p<0.05).

### Effect of the intervention on confidence
At baseline there was no difference in the percentage of participants in the intervention and control sites who were confident or very confident in counselling mothers on HIV/infant feeding (online supplemental table 2). However, at follow-up HWs from intervention sites were significantly more confident in counselling women living with HIV about HIV and infant feeding, returning to school/work, continuing breast feeding for 2 years, assessing ART adherence in women living with HIV, and advising women living with HIV about breast feeding with cracked nipples (online supplemental table 2). Confidence had not shifted about how to stop breast feeding, identifying when a mother is not ART-adherent and managing poor adherence, advising on formula feeding and counselling that a shorter breastfeeding duration is better than no breast feeding. The mean confidence score at follow-up was significantly higher in the intervention compared with the comparison sites at follow-up (p=0.05) (online supplemental table 2). Methods 1 and 2 yielded the same results, except for the effect of baseline confidence score. Thus, in table 5 below, we only show results for Methods 1 and 3. Controlling for other variables, postintervention confidence was significantly better in

intervention, compared with comparison, sites; however this was only statistically significant under Method 1. Our analysis demonstrated that, controlling for other factors, being a trained health professional significantly increased confidence score by 3.1 (Method 1) or 3.7 (Method 3). Additionally, Method 3 demonstrated that, controlling for other factors, working for less than 5 years significantly reduced the confidence score.

### Dose-response analysis
We also conducted a dose-response analysis to assess whether or not the mentored HWs responded differently according to the number of workshops attended (0, 1 or 2, and 3). Even though postintervention attitude and confidence scores as well as their increases were higher in the higher workshop attendance participants, there was no statistically significant dose-response effect (p value>0.05, data not shown)

### DISCUSSION
We demonstrate that a participatory, side-by-side, team-based mentoring approach to disseminating updated HIV and infant feeding guidelines was associated with an improvement in HWs' attitudes. when controlling for other factors. There was also a significant improvement in mean knowledge score between intervention and control

**Table 5** Adjusted effect of the intervention on health worker confidence scores, using different multivariable analysis methods (effect estimate and 95% CI)

| Variable | Method 1 | | Method 3 | |
|---|---|---|---|---|
| | Effect estimate | 95% CI | Effect estimate | 95% CI |
| Confidence Score at baseline | 0.4 | 0.3 to 0.6[*] | N/A | N/A |
| Intervention | 2.4 | 0.3 to 4.5[*] | 1.5 | −2.2 to 5.1 |
| Follow-up time | N/A | N/A | 0.5 | −1.5 to 2.5 |
| Cadre of heath professional versus community level | | | | |
| Trained health professional | 3.1 | 0.3 to 5.9[*] | 3.7 | 1.5 to 5.9[*] |
| Enrolled nurse | −0.8 | −4.3 to 2.7 | −0.7 | −3.1 to 1.6 |
| Ugu District versus Tshwane District | 0.00 | −2.1 to 2.1 | −1. | −3.2 to 1.2 |
| Age category versus 23–35 years | | | | |
| 36–41 years | −1.0 | −3.7 to 1.6 | −0.1 | −2.7 to 2.5 |
| 42–46 years | 0.3 | −2.9 to 3.4 | 0.4 | −1.2 to 2.9 |
| 47–54 years | 1.4 | −0.7 to 3.5 | −1.3 | −3.4 to 0.8 |
| Over 54 years | −2.5 | −5.7 to 0.7 | −0.9 | −4.0 to 2.2 |
| Work experience <5 years vs ≥5 years | −0.5 | −3.4 to 2.4 | −1.9 | −3.7 to −0.2[*] |
| Received training or information at work about the revised policy | 0.05 | −1.5 to 1.6 | 1.7 | −0.3 to 3.6 |
| Received any training about managing common breastfeeding problems? | −0.6 | −3.2 to 2.1 | 1.8 | −0.5 to 4.1 |
| Ever received any training about how to assess and support antiretroviral therapy adherence for HIV-positive women? | 0.8 | −2.1 to 3.7 | 5.7 | 3.5 to 7.9 |

All analyses are adjusted for clustering.
*p<0.005.
N/A, not applicable.

sites at follow-up. However, we were not successful in shifting knowledge and attitudes about mixed feeding (breast milk and formula milk) and HWs at the end of the study were not confident in advising that a shorter duration of breast feeding is better than no breast feeding at all. This demonstrates the success of at least 15 years of frequent publicity about the dangers of mixed feeding in the context of HIV and no ART, exacerbated by the fact that the two seminal papers on feeding practices and HIV were led by South African researchers.[35 36] Concerted communication efforts are needed to highlight the acceptability of mixed feeding in the context of ART and maternal viral load suppression to facilitate a shift in knowledge about mixed feeding. Although some individual attitude and confidence items did not change, or only changed marginally, the overall analyses demonstrated an improvement in follow-up attitude and confidence scores. However, confidence in the intervention group was still low and HWs performed poorly on some of the more difficult confidence questions such as confidence with counselling when a mother is not ART-adherent, managing high viral loads during breast feeding, explaining HIV transmission risks to a mother with a high viral load, assisting women living with HIV to safely formula feed and advising that some breast feeding is better than no breast feeding. The complexity of changing HWs' attitudes and confidence towards breast feeding has been documented repeatedly in many African settings including South Africa.[11 25 37–39] We hypothesise that poor performance on some of the individual items or on the overall confidence score may be attributed to the short duration of the intervention. An alternative hypothesis is that HWs' low confidence around topics like non-adherence and high viral load reflect more complex dynamics that are not easily addressed through counselling/mentoring interventions.[39] In fact a study from South Africa demonstrated how HWs' personal beliefs affect their ability to provide supportive counselling.[11]

There is evidence that in-service training, supervision and follow-up improves the knowledge, skills and practices of HWs managing childhood undernutrition, and can improve HW job satisfaction and motivation, but no data exist on how to improve HW knowledge, skills and confidence in the tricky area of HIV and infant feeding.[40–42] For training/supervision interventions, implementation challenges include inadequately trained supervisors or shortages of supervisors, inappropriate job aids for follow-up, and poor alignment between community views/practice and health programmes.[43] Our approach attempted to circumvent these challenges by using a low technology mentorship model for skills development at clinic level. At the outset of the intervention we acknowledged that HWs were members of their community: we discussed their fears and beliefs, and then introduced facts and evidence to extend their knowledge, change their attitudes, and increase their confidence

to implement updated guidelines on HIV and infant feeding. Thus, we aimed to change inherent, deep-seated beliefs and attitudes that are sustained in the absence of outside supervision.

We used a side-by-side mentorship approach, as reviewed by Schwerdtle *et al*, to conduct team-based mentoring to empower HWs.[44] A team-based approach allowed collaborative learning between different cadres of HWs, facilitating any future change in practice. In accordance with Dee Fink's theory, a participatory mentorship approach allows participants to develop foundational knowledge, apply skills, integrate ideas, develop new feelings/interests and values, and learn how to learn.[26] Our experience suggests that such an approach allowed discussion of participants' attitudes towards performing the behaviour, beliefs about whether critical, important people will approve of the behaviour (subjective norms), and about their likelihood of successfully implementing the behaviour.[28 29] Our findings corroborate a scoping review which demonstrated that mentorship improves certain quality of care outcomes;[44] in our study it improved knowledge, attitudes and confidence. However, only four studies were included in this scoping review, and the nature of the mentorship varied from videoconferencing to monthly, 6-weekly or annual visits interspersed with other contact forums, conducted over 1 day to an entire week. A list of desirable features of mentorship interventions include at least one dedicated mentor per facility, ensuring an adequate mentor:mentee ratio so that all staff can be supported, forming meaningful relationships between mentors and mentees, ensuring cultural congruency between mentee and mentor, and using mentors for protocol-driven programmes such as Integrated Management of Childhood Illness Strategy (IMCI) or HIV.[44] Our intervention related to HIV and infant feeding guidelines, was low cost and low technology (one mentor working with pen, flip chart and paper in the health facility), and was implemented by a dedicated mentor from the same cultural background as the mentees. She provided onsite support during the workshops, which lasted approximately 1 hour, and additional support through a WhatsApp messaging group.

There is an ongoing heath worker crisis in resource-limited settings, including maldistribution of staff, an imbalance in skills mix, increasingly complicated health programmes and complicated sociocultural-political-economic environments. Against this background, questions arise about the feasibility of an on-site mentorship approach to guideline dissemination among HWs, and an on-site peer-peer mentorship approach between women living with HIV to supporting mothers with infant feeding. In this study we chose to focus specifically on an on-site mentorship approach to guideline dissemination among HWs. We argue that strengthening investment in on-site mentorship rather than off-site training, may be a cost-saving approach. In our setting, all clinics receive regular visits from district PHC supervisors, but these mostly focus on administration and clinic management matters. These supervisors, as well as existing district PHC trainers, could be capacitated to provide clinical mentoring for HWs in the clinics they oversee. Our model of team-based learning and mentoring can be used for on-site mentoring, and avoids accommodation and travel costs, and absence from work that off-site training requires.

Our study had several limitations: We purposively selected districts for inclusion. We could not control for previous breastfeeding experiences of HWs as we did not gather these data. The study tools were piloted before finalisation, but no factor analyses or validation exercises were conducted. The follow-up evaluation was conducted 3 months after the intervention; thus, we were only able to measure short-term benefits. Additionally, we did not co-design the intervention with women living with HIV, did not measure the effect of improved knowledge, attitudes and confidence on HWs' counselling practices and on mothers' infant feeding practices, and could not tease out whether the relationship between number of workshops and outcomes was due to staff motivation (more motivated staff attended more workshops) or the workshops themselves. Co-designing the intervention with mothers living with HIV may have resulted in a different intervention and results; this needs to be considered in future work. Our study's strengths are that the design was quasi-experimental, measuring knowledge, and attitudes and confidence. Additionally, results are robust as three different analytical methods yielded congruent results.

## Conclusion

We demonstrated improved knowledge, attitudes and confidence of HWs following a participatory mentorship approach to HIV and infant feeding guideline dissemination compared with a standard approach. More research is needed to better understand how to change HWs' counselling practices, and whether this changes mothers' feeding practices.

**Author affiliations**
[1]Health Systems Research Unit, South African Medical Research Council, Tygerberg, South Africa
[2]Department of Paediatrics, University of Pretoria, Pretoria, South Africa
[3]HIV Prevention Research Unit, South African Medical Research Council, Tygerberg, South Africa
[4]School of Public Health, University of the Western Cape, Cape Town, South Africa
[5]School of Public Health, University of the Witwatersrand, Johannesburg, South Africa
[6]Biostatistics Unit, South African Medical Research Council, Tygerberg, South Africa
[7]Department of Statistics, University of Pretoria, Pretoria, South Africa
[8]Centre for Rural Health, University of KwaZulu-Natal, Durban, South Africa
[9]School of Education, University of KwaZulu Natal, Pietermaritzburg, South Africa
[10]Centre for International Health, University of Bergen, Bergen, Norway
[11]Gauteng Department of Health, Tshwane District Health Services, Pretoria, South Africa
[12]Research Centre for Maternal, Fetal, Newborn and Child Health Care Strategies, University of Pretoria, Pretoria, South Africa
[13]Maternal and Infant Health Care Strategies Research Unit, South African Medical Research Council, Tygerberg, South Africa
[14]Division of Human Nutrition and Department of Paediatrics and Child Health, Stellenbosch University, Cape Town, South Africa
[15]Burden of Disease Research Unit, South African Medical Research Council, Tygerberg, Switzerland

[16]Department of Maternal, Newborn, Child and Adolescent Health, World Health Organization, Geneva, Switzerland
[17]Department of Neonatology, Faculty of Health Sciences, University of Cape Town, Cape Town, South Africa
[18]Mowbray Maternity Hospital, Cape Town, South Africa
[19]Department of Paediatrics and Child Health, University of Cape Town, Cape Town, South Africa
[20]National Department of Health, Pretoria, South Africa

**Contributors** AG: Study conceptualisation and tool development, protocol writing including intervention development, oversight of sampling and field work, writing of the first draft of this manuscript, receiving and incorporating coauthor comments, finalisation of the paper. TD: Study conceptualisation and tool development, protocol writing including intervention development, set up the sample frame and sampling, contributed to the manuscript, reviewed and approved the final version of the manuscript. SM: Led the statistical components of the protocol; provided overall oversight on the statistical analysis, contributed to the manuscript, reviewed and approved the final version of the manuscript. TN: Performed the work on the statistical components of the protocol, under SM's guidance; provided data analysis under SM's guidance, contributed to the manuscript, reviewed and approved the final version of the manuscript. LH: Contributed to study conceptualisation and tool development, protocol writing including intervention development; was overall Project Manager; established, managed and cleaned the database; contributed to the manuscript, reviewed and approved the final version of the manuscript. VMJ: Provided guidance on intervention development. Contributed to the manuscript, reviewed and approved the final version of the manuscript. IMSE: Contributed to study conceptualisation and tool development, protocol writing including intervention development; contributed to the manuscript, reviewed and approved the final version of the manuscript. UF: Contributed to study conceptualisation, assisted with district-level buy-in in Tshwane District, provided information on routine dissemination of updated infant feeding guidelines; contributed to the manuscript, reviewed and approved the final version of the manuscript. AD: Contributed to study conceptualisation and tool development; contributed to the manuscript, reviewed and approved the final version of the manuscript. NR: Contributed to study conceptualisation; contributed to the manuscript, reviewed and approved the final version of the manuscript. MK: Contributed to study conceptualisation and tool development; contributed to the manuscript, reviewed and approved the final version of the manuscript. DS: Contributed to study conceptualisation and tool development; contributed to the manuscript, reviewed and approved the final version of the manuscript. SK: Contributed to study conceptualisation, assisted with national-level buy-in, provided information on - routine dissemination of updated infant feeding guidelines; contributed to the manuscript, reviewed and approved the final version of the manuscript. TT: Contributed to study conceptualisation and tool development; contributed to the manuscript, reviewed and approved the final version of the manuscript. CH: Study conceptualisation, protocol writing including intervention development, high-level oversight of study implementation, contributed to the manuscript, reviewed and approved the final version of the manuscript.

**Funding** This study was funded by the WHO (Ref: 2017/712509-0). TD receives a from the National Research Foundation, South Africa. AG's and TD's time was funded by the South African Medical Research Council. The publication of paper was funded by the South African Medical Research Council.

**Map disclaimer** The depiction of boundaries on this map does not imply the expression of any opinion whatsoever on the part of *BMJ* (or any member of its group) concerning the legal status of any country, territory, jurisdiction or area or of its authorities. This map is provided without any warranty of any kind, either express or implied.

**Competing interests** None declared.

**Patient consent for publication** Not required.

**Ethics approval** Ethics approval was obtained from the South African Medical Research Council (EC028-9/2016), the University of KwaZulu-Natal (RECIP348/17) and the WHO Ethics Review Committee (ERC0002833).

**Provenance and peer review** Not commissioned; externally peer reviewed.

**Data availability statement** Data can be obtained by emailing the corresponding author, and upon reasonable request.

**ORCID iDs**
Ameena Goga http://orcid.org/0000-0002-2394-6486
Ingunn M S Engebretsen http://orcid.org/0000-0001-5852-3611

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
