## [Reviewer comments · BMJ Open]

ARTICLE DETAILS

TITLE (PROVISIONAL)	Translating new evidence into clinical practice: A quasi-experimental controlled before-after study evaluating the effect of a novel outreach mentoring approach on knowledge, attitudes and confidence of health workers providing HIV and infant feeding counselling in South Africa
AUTHORS	Goga, Ameena; Doherty, Tanya; Manda, Samuel; Nkwenika, Tshifhiwa; Haskins, Lyn; John, Vaughn; Engebretsen, Ingunn; Feucht, Ute; Dhansay, Ali; Rollins, Nigel; Kroon, Max; Sanders, David; Kauchali, Shuaib; Tylleskär, Thorkild; Horwood, Christiane

VERSION 1 – REVIEW

REVIEWER	Sara Jewett Nieuwoudt School of Public Health, University of the Witwatersrand, Johannesburg, South Africa
REVIEW RETURNED	19-Dec-2019

GENERAL COMMENTS	I enjoyed reading about this clinic-based intervention to support health workers to engage with updated infant feeding guidelines. The study design was robust and the article is generally well written. Insight on this topic is needed in the South African context. The following review is designed to further strengthen this piece in hopes that it will be published. I begin with substantive issues related to the content for the authors to consider, followed by minor edits. As continuous numbering wasn't applied, please note the page numbers. Content Comments ABSTRACT/STRENGTHS/LIMITATIONS Specify the time period over which data were collected (See p.1, line 14 and p.5, lines 15-16). This is important for making claims about dissemination of the policy changes. The limitations suggest that you were seeking to change “actual infant feeding practices” while the theory and constructs indicate you were focusing on HW infant feeding counselling practices (also not measured). This is more in line with the study population of interest (p.1, line 49). Similarly, clarify what the study was intending to change knowledge, attitudes and confidence about, e.g. counselling based on the latest evidence, on p.4, line 10 in the Background. BACKGROUND
---

Add a reference to the frequent changes to infant feeding guidelines, e.g. Jackson et al, 2019 in BMJ Open or Nieuwoudt et al, 2019 in PLOS One. Alternatively, reference the actual guidelines that changed over the past decade. Poor dissemination of past/changing guidelines to HWs has also been written about in the South African literature and could be referenced. (p.3, line 23)

Add primary reference for the PMTCT guidelines (p3, line 40) and reference the policy shifts in South Africa that resulted. There is no mention of the guideline changes after the Tshwane Declaration, which was arguably one of the more dramatic shifts in PMTCT counselling for HIV exposed infants in the public health setting. On p4, lines 20-21, the wording suggests that the 2017 change was the only (or most significant) that the two provinces have experienced, which may not be the case.

METHODS

Specify and reference the technique(s) used to adjust for clustering on p.5, line 6 and/or p.6, line 52.

The authors need to engage with the possibility that the Horwood et al article will not be published by the time this is accepted (see p5, line 17 and line 55). If this happens, more description of the intervention will be needed.

The question of whether and how the 2017 circular was disseminated to clinics (p5, lines 46-8) through routine systems is important and should feature more strongly in analysis of intervention results, beyond Supplemental Figure 1. Specific suggestions are made in results section.

Add references to existing tools you used/adapted and note whether or not they are validated (p. 6, line 8 or in section starting on line 23). Please indicate if the study team conducted any factor analysis or validation exercises on the three scales? In reviewing the tables, I am concerned that several questions are double-barreled or overly complex so as to distort findings. For example, the first part of the knowledge question about sterilization of bottles is True and the second part is False. The low levels of correct answers may relate to the nature of the way the question was framed. This is a potential limitation that should be acknowledged.

RESULTS

p.7, lines 30-9: Were the three statistically significant differences in training between comparison and intervention sites explored and/or integrated into the models measuring differences in attitudes and confidence (Tables 4 & 5)? If so, this could strengthen the argument for an intervention effect. If not, please explain their exclusion or consider adding this to your analysis.

p.9, from line 21. For knowledge interpretation, it may be worth highlighting the questions in Table 3 that contradicted prior PMTCT guidelines to support interpretation and deepen discussion.

What was the rationale for not analysing knowledge using the same robust statistical techniques to those applied for attitudes and confidence?

Supplementary Tables 1 and 2: The p-values for individual items presented in the table seem to be incorrect or inconsistent. For example, in line 16 of the Table 1, 90% vs. 83% has a p-value of 0.14, while on line 19, 88.6% vs. 88.4% has a p-value of 0.06. I recommend that these are checked and that the results (p.12, lines 5-9 for Supp Table 1 and p.13, lines 5-6 for Supp Table 2) are updated if needed. This issue was noted for both tables.

The “three analyses” conducted to create Tables 4 and 5 are not specified. Particularly given that only two models are presented (see p.12, line 9 and 19; p.13, line 9), the test(s) used to generate the first model in both tables needs to be specified and differentiated. On p.12, line 20, “ANCOVA or linear regression analyses” are mentioned. One or the other should be specified as the basis for the tables (or explain that they had identical results if that was the case).

DISCUSSION

While logically presented, the discussion ventures beyond the study findings with several arguments with limited empirical evidence. I recommend that the authors identify studies from the global literature that can bolster or support the following arguments that are made:

1. That improvements in knowledge, attitudes and/or confidence will result in HWs improving their counselling practices and that improved HW counselling can lead to improved infant feeding practices (p.14, lines 24-26)
2. The intervention is a sustainable model (p.14, line 38). [NB: My understanding is that a dedicated mentor/facilitator was required to run a series of workshops across clinics. Without a clear dose response, is this sufficient evidence for DOH to invest in mentors at scale?]
3. On-site mentorship is more cost-effective than off-site (assuming DOH is willing to invest in any mentorship scheme) (p.15, lines 25-30)

LIMITATIONS

p.15, line 37: Are you referring to the personal breastfeeding experiences of HWs or their previous experiences of providing infant feeding counselling? This is unclear and both limitations could be argued

The validity of the scales used to measure the concepts of knowledge, attitudes and confidence should be addressed in the limitations or explained in greater detail in methods.

MINOR EDITS

p.2, line 25: Ensure percentages are all presented with the same level of specificity (decimals)

p.2, line 31 and p3, line 52: Move apostrophe in HW's to HWs'

p.3, line 35: For WHO, Organization is spelled with a 'z'

p.3, line 48: Consider starting a new paragraph with the sentence beginning, “Many studies on the uptake...” as this is engaging with new ideas

p.3, line 52: Only one study is referenced while the sentence refers to “some”

p.4, line 20: Revise to policy change (not changed)

	p.6, line 39: Is this second sub-heading necessary? This information still falls under data analysis. p.7, line 17: Add a decimal place to 50% to standardize measure specificity p.9, lines 48 and 51: use conventional $p < 0.001$ rather than $p = 0.0000$ Table 1. Standardize the number of decimal places (see line 21 [40.0] and 28 [0.20]) p.12, line 23: Consider using the word “better” to describe attitude rather than “higher” for easier interpretation Table 3, p.11. The = sign used before some statements is confusing and I thought it was a typo at first. I suggest rather using a more common signifier, like **, to indicate that the reader should look for the explanation below the table. p.12, line 9: Remove the comma before (Supplementary Table 1) Supplementary Table 1: There is a * in the p-values columns and again later in the table linked to attitude scores. Ensure that the test used to calculate p-values is described and that the second * is updated to ** or another symbol to differentiate. Tables 4 & 5: For both tables, shouldn't the Intervention diff-in-diff analysis be in bold like other significant findings for consistency. Also, check if the * description below the table is meant to read $p < 0.05$ rather than $p < 0.005$ p.16, line 13: Comma isn't needed in phrase “prevalence is high, is low” p.16, line 28: did you mean to have \leq next to p or simply $p < 0.001$?
--	---

REVIEWER	Anna Helova The University of Alabama at Birmingham, Alabama, United States
REVIEW RETURNED	12-Apr-2020

GENERAL COMMENTS	Well-written paper. Revisions and suggestions: Table 1. It would be good to show distribution of each characteristic per province. Table 2. Show numbers for Tshwane and Ugu. Table 3. Show within the group differences in addition to between the group differences. Strengths and limitations: include something on the differences between provinces, and that the knowledge score sometimes increased at comparison sites. Formatting/grammar: extra space (page 3, line 46), delete 'of' (page 3, line 53), incorrect punctuation (page 15, line 46).
--

REVIEWER	Alice Welbourn Salamander Trust, UK Supporter of 4M Mentor Mother Network CIC, UK. Woman living with HIV. Member of guideline development group, WHO 2017 consolidated guideline on SRHR of women living with HIV; coauthor of various related and follow-up documents.
REVIEW RETURNED	16-Apr-2020

GENERAL COMMENTS	This manuscript is tantalising in that the approach to participatory mentoring of the health workers concerned is to be congratulated. However the huge elephant in the room is the apparent total lack of involvement of women living with HIV, who are supposed to be the beneficiaries of this training, or their values and preferences in the
---

process; and no commentary on this. Given the growing body of research, practice and guidance in this complex area, this is a considerable and disappointing omission. For example, there is clear guidance in the WHO 2017 Consolidated Guideline on the SRHR of women living with HIV that women living with HIV should be meaningfully involved in all research that affects their lives (section 6.2.1 especially); and that “peer support, provided by, with, and for women living with HIV, should be included in HIV care” (GPS A.1). This guideline was published before the research baseline was conducted in August 2017. There is also an RCT (Richter et al 2014) that describes the advantage of a peer mentor mother process - yet none seemed to be in place - or was mentioned - in this manuscript. Documents published by WHO more recently have clearly described the huge added value of the meaningful involvement of women living with HIV (see eg WHO 2019a (<https://www.who.int/reproductivehealth/publications/srhr-women-hiv-implementation/en/>) and WHO 2019b (<https://apps.who.int/iris/bitstream/handle/10665/330034/WHO-RHR-17.33-eng.pdf?ua=1>)). Amongst other benefits, the use of women living with HIV to inform health workers directly of the many and complex issues they face, including in relation to infant feeding, have been documented. It is also important for health workers to be trained in trauma-informed care, so that they are aware of the full range of potential barriers to access to care and treatment experienced by women living with HIV. These are very likely to include violence against women at home and in health care settings, and concomitant anxiety, depression and other mental health challenges (see eg Orza et al JIAS 2015a and 2015b, Orza et al JIAS 2017, Desliets et al J Assoc Nurses AIDS Care 2020). More recent studies in the UK amongst migrant communities have clearly shown the benefits of peer mentor mother support networks run by and for women living with HIV (ie not just by external NGOs or health service providers) (eg Cameron et al 2018 https://salamandertrust.net/wpcontent/uploads/2018/04/4M_LSHTM_BHIVA-PosterFINAL2018.pdf ; and Hay et al 2020 <https://www.tandfonline.com/doi/full/10.1080/09540121.2020.1739220>). It is also tantalising that the study was considered to be complete before there was a chance for the women concerned to comment on whether or not the support they were receiving had improved. There is often a big gap between what women actually want, or prioritise, and what health service providers think they need. See eg 2 documents to support this: a) using the UNAIDS-commissioned ALIV[H]E framework in MENA region (p15 especially) https://salamandertrust.net/wpcontent/uploads/2017/11/ALIVHE_in_Action_FINAL_Salamander_et_al_March2019.pdf and b) Salamander Trust, PIPE and UNYPA 2018 - see eg images on p4 https://issuu.com/salamandertrust.net/docs/20180308_4m_advocacybrieflowresfina).

In addition to these overall comments, here are some more specific ones for your consideration:
LANGUAGE - please be mindful of avoiding use of stigmatising and derogatory language throughout. See eg Dilmitis et al 2010 JIAS; and UNAIDS terminology guide 2015.
BEHAVIOUR CHANGE. Refs 14, 15 and 16 These are all very old behaviour change models. While the Fink has merits, the linear behaviour change model is definitely very simplistic. See also eg

	https://academic.oup.com/heapro/article/34/3/616/4951539?guestAccessKey=c0cf0f69-57a3-4a35-9df9-091baa1e2ee0 HEALTH STAFF - were any of the trainee health staff also living with HIV? The manuscript appears to assume not. This is a potential missed opportunity for shared insights and learning. See eg Odetoyinbo et al JIAS. 2009 https://www.ncbi.nlm.nih.gov/pmc/articles/PMC2664321/ TABLE 3 - please specifically reference the source of the true information in this table for ease of access for the reader.
--	--

VERSION 1 – AUTHOR RESPONSE

Reviewer(s)' Comments to Author:

Reviewer: 1

Specify the time period over which data were collected (See p.1, line 14 and p.5, lines 15-16). This is important for making claims about dissemination of the policy changes.
This has now been added.

The limitations suggest that you were seeking to change “actual infant feeding practices” while the theory and constructs indicate you were focusing on HW infant feeding counselling practices (also not measured). This is more in line with the study population of interest (p.1, line 49). Similarly, clarify what the study was intending to change knowledge, attitudes and confidence about, e.g. counselling based on the latest evidence, on p.4, line 10 in the Background.

Thank you for this comment - this has now been clarified. We were not seeking to measure changes in mothers’ practices: we sought to improve health care workers’ knowledge of, attitudes towards and confidence with counselling on HIV and infant feeding. However, we conducted 12 focus groups amongst mothers attending these facilities approximately 3 months after the health worker training. These findings have been published - see Doherty T, Horwood C, Haskins L, Magasana V, Goga A, Feucht U, et al. Breastfeeding advice for reality: Women's perspectives on primary care support in South Africa. *Maternal & Child Nutrition* 2020; 16(1):e12877 DOI: 10.1111/mcn.12877.
In summary the qualitative results demonstrate the need to strengthen the confidence of health care providers and extend breastfeeding support into the community.

BACKGROUND

Add a reference to the frequent changes to infant feeding guidelines, e.g. Jackson et al, 2019 in *BMJ Open* or Nieuwoudt et al, 2019 in *PLOS One*. Alternatively, reference the actual guidelines that changed over the past decade. Poor dissemination of past/changing guidelines to HWs has also been written about in the South African literature and could be referenced. (p.3, line 23)

Add primary reference for the PMTCT guidelines (p3, line 40) and reference the policy shifts in South Africa that resulted. There is no mention of the guideline changes after the Tshwane Declaration, which was arguably one of the more dramatic shifts in PMTCT counselling for HIV exposed infants in the public health setting. On p4, lines 20-21, the wording suggests that the 2017 change was the only (or most significant) that the two provinces have experienced, which may not be the case. Additional references have been added.

Our statement says that:

'In June 2017, these recommendations were adopted in South Africa, thus necessitating updates for health workers.' It does not specifically say that only these two provinces were affected.

METHODS

Specify and reference the technique(s) used to adjust for clustering on p.5, line 6 and/or p.6, line 52. A reference has now been added to provide clarity.

The authors need to engage with the possibility that the Horwood et al article will not be published by the time this is accepted (see p5, line 17 and line 55). If this happens, more description of the intervention will be needed.

The paper has been published and added as a reference.

The question of whether and how the 2017 circular was disseminated to clinics (p5, lines 46-8) through routine systems is important and should feature more strongly in analysis of intervention results, beyond Supplemental Figure 1. Specific suggestions are made in results section. The district office sent the circular via e-mail and whats app to all health facilities. This has now been added on pg. 6. The study team documented that the June 2017 circular issued by the National Department of Health, informing health facilities of the change in Infant and Young Child feeding policy, was disseminated to comparison clinics as an announcement via e-mail and other electronic communication as well as during meetings or trainings. We documented that in Tshwane, 15 of the 18 clinics had received the circular; 11 via e-mail and three at a meeting. In Ugu nine of 17 clinics had received the circular; 8 received it via hand delivery and one via e-mail.

Add references to existing tools you used/adapted and note whether or not they are validated (p. 6, line 8 or in section starting on line 23). Please indicate if the study team conducted any factor analysis or validation exercises on the three scales?

References have been added. Factor analyses were not conducted on the tools and they were not validated; however they were reviewed by an education specialist and were piloted amongst health care providers. This is explained in the intervention paper that has been published: Christiane Horwood , Lyn Haskins, Ameena Goga, Tanya Doherty, Vaughn John, Ingunn M.S. , Engebretsen, Ute Feucht, Nigel Rollins, Max Kroon, David Sanders, Thorkild Tylleskar. An educational intervention to update health workers about HIV and infant feeding. *Maternal & Child Nutrition*.

<https://doi.org/10.1111/mcn.12922>

It is also explained more clearly in the data collection section of this paper.

In reviewing the tables, I am concerned that several questions are double-barreled or overly complex so as distort findings. For example, the first part of the knowledge question about sterilization of bottles is True and the second part is False. The low levels of correct answers may relate to the nature of the way the question was framed. This is a potential limitation that should be acknowledged. Thanks for this – it has been added as a limitation.

RESULTS

p.7, lines 30-9: Were the three statistically significant differences in training between comparison and intervention sites explored and/or integrated into the models measuring differences in attitudes and confidence (Tables 4 & 5)? If so, this could strengthen the argument for an intervention effect. If not, please explain their exclusion or consider adding this to your analysis.

Thank you for raising this. The analysis has been re-done to control for these findings and Tables 4 and 5 have been updated.

p.9, from line 21. For knowledge interpretation, it may be worth highlighting the questions in Table 3 that contradicted prior PMTCT guidelines to support interpretation and deepen discussion.

The questions really measuring the main change in the 2017 guidelines are:

“Continued breastfeeding for 2 years is the recommended infant method in SA for ALL children, regardless of mother’s HIV status (True)” and

“In South Africa, HIV-infected women who are breastfeeding should be supported to adhere to antiretroviral treatment and should introduce complementary foods around 6 months and be supported to continue breastfeeding for at least two years. (True)”

For the first question in the intervention group there was a 36% improvement in knowledge whilst in the comparison group there was a 13% increase in knowledge.

For the second question in the intervention group there was a 15% increase in correct knowledge for this question while for the comparison group knowledge decreased from 89-81%.

So, these findings supports the conclusion that knowledge increased and especially for the main change in the guideline i.e. duration of breastfeeding.

These findings have been flagged in the results, in the table and in the discussion.

What was the rationale for not analyzing knowledge using the same robust statistical techniques to those applied for attitudes and confidence?

We were more interested in understanding changes in attitudes and confidence, as knowledge can easily be improved through studying/ training, whilst attitude and confidence are more difficult to change; hence the more advance analyses for attitudes and confidence.

Supplementary Tables 1 and 2: The p-values for individual items presented in the table seem to be incorrect or inconsistent. For example, in line 16 of the Table 1, 90% vs. 83% has a p-value of 0.14, while on line 19, 88.6% vs. 88.4% has a p-value of 0.06. I recommend that these are checked and that the results (p.12, lines 5-9 for Supp Table 1 and p.13, lines 5-6 for Supp Table 2) are updated if needed. This issue was noted for both tables.

Apologies for this and thank you for pointing this out – there were incorrect transcriptions in this table These have now been rectified.

The “three analyses” conducted to create Tables 4 and 5 are not specified. Particularly given that only two models are presented (see p.12, line 9 and 19; p.13, line 9), the test(s) used to generate the first model in both tables needs to be specified and differentiated. On p.12, line 20, “ANCOVA or linear regression analyses” are mentioned. One or the other should be specified as the basis for the tables (or explain that they had identical results if that was the case).

More details have now been added under Methodology to explain the different models

DISCUSSION

While logically presented, the discussion ventures beyond the study findings with several arguments with limited empirical evidence. I recommend that the authors identify studies from the global literature that can bolster or support the following arguments that are made:

1. That improvements in knowledge, attitudes and/or confidence will result in HWs improving their counselling practices and that improved HW counselling can lead to improved infant feeding practices (p.14, lines 24-26)
2. The intervention is a sustainable model (p.14, line 38). [NB: My understanding is that a dedicated mentor/facilitator was required to run a series of workshops across clinics. Without a clear dose response, is this sufficient evidence for DOH to invest in mentors at scale?]

3. On-site mentorship is more cost-effective than off-site (assuming DOH is willing to invest in any mentorship scheme) (p.15, lines 25-30)

The discussions has been reworked to address how the findings are framed and interpreted.

LIMITATIONS

p.15, line 37: Are you referring to the personal breastfeeding experiences of HWs or their previous experiences of providing infant feeding counselling? This is unclear and both limitations could be argued

We are referring to the health care workers own personal breastfeeding experiences and have clarified in the text.

The validity of the scales used to measure the concepts of knowledge, attitudes and confidence should be addressed in the limitations or explained in greater detail in methods.

The limitations have been reworded and clarified. A point has been added about piloting versus validation of the tools.

MINOR EDITS

p.2, line 25: Ensure percentages are all presented with the same level of specificity (decimals) All percentages are reported to one decimal place, except for p-values

p.2, line 31 and p3, line 52: Move apostrophe in HW's to HWs' This has been changed

p.3, line 35: For WHO, Organization is spelled with a 'z' Done

p.3, line 48: Consider starting a new paragraph with the sentence beginning, "Many studies on the uptake..." as this is engaging with new ideas Done

p.3, line 52: Only one study is referenced while the sentence refers to "some" Done

p.4, line 20: Revise to policy change (not changed) Done

p.6, line 39: Is this second sub-heading necessary? This information still falls under data analysis. This has been deleted

p.7, line 17: Add a decimal place to 50% to standardize measure specificity Done

p.9, lines 48 and 51: use conventional $p < 0.001$ rather than $p = 0.0000$ Done

Table 1. Standardize the number of decimal places (see line 21 [40.0] and 28 [0.20]) Done

p.12, line 23: Consider using the word "better" to describe attitude rather than "higher" for easier interpretation Done

Table 3, p.11. The = sign used before some statements is confusing and I thought it was a typo at first. I suggest rather using a more common signifier, like **, to indicate that the reader should look for the explanation below the table. Done

p12, line 9: Remove the comma before (Supplementary Table 1) Done

Supplementary Table 1: There is a * in the p-values columns and again later in the table linked to attitude scores. Ensure that the test used to calculate p-values is described and that the second * is updated to ** or another symbol to differentiate. This has been corrected

Tables 4 & 5: For both tables, shouldn't the Intervention diff-in-diff analysis be in bold like other significant findings for consistency. Also, check if the * description below the table is meant to read $p < 0.05$ rather than $p < 0.005$ Done

p.16, line 13: Comma isn't needed in phrase "prevalence is high, is low" This summary is not needed and so has been deleted

p.16, line 28: did you mean to have \leq next to p or simply $p < 0.001$? This summary is not needed and so has been deleted

Reviewer: 2

Please leave your comments for the authors below
Well-written paper.

Revisions and suggestions:

Table 1. It would be good to show distribution of each characteristic per province.

Table 2. Show numbers for Tshwane and Ugu.

Table 3. Show within the group differences in addition to between the group differences.

The study was not designed to show differences by site and province; thus we are not comfortable reporting the data by the two sites. However, we have reviewed the data by site, and decided to combine the data because there were few differences between the sites.

Strengths and limitations: include something on the differences between provinces, and that the knowledge score sometimes increased at comparison sites. The study and sample size was not designed to show differences by site and province; thus we are not comfortable reporting the data by the two sites.

Formatting/grammar: extra space (page 3, line 46), delete 'of' (page 3, line 53), incorrect punctuation (page 15, line 46).

These have been corrected.

Reviewer: 3

Please leave your comments for the authors below

This manuscript is tantalising in that the approach to participatory mentoring of the health workers concerned is to be congratulated. However the huge elephant in the room is the apparent total lack of involvement of women living with HIV, who are supposed to be the beneficiaries of this training, or their values and preferences in the process; and no commentary on this. Given the growing body of research, practice and guidance in this complex area, this is a considerable and disappointing omission. For example, there is clear guidance in the WHO 2017 Consolidated Guideline on the SRHR of women living with HIV that women living with HIV should be meaningfully involved in all research that affects their lives (section 6.2.1 especially); and that “peer support, provided by, with, and for women living with HIV, should be included in HIV care” (GPS A.1). This guideline was published before the research baseline was conducted in August 2017. There is also an RCT (Richter et al 2014) that describes the advantage of a peer mentor mother process - yet none seemed to be in place - or was mentioned - in this manuscript. Documents published by WHO more recently have clearly described the huge added value of the meaningful involvement of women living with HIV (see eg WHO 2019a (<https://www.who.int/reproductivehealth/publications/srhr-women-hiv-implementation/en/>) and WHO 2019b (<https://apps.who.int/iris/bitstream/handle/10665/330034/WHO-RHR-17.33-eng.pdf?ua=1>)). Amongst other benefits, the use of women living with HIV to inform health workers directly of the many and complex issues they face, including in relation to infant feeding, have been documented. It is also important for health workers to be trained in trauma-informed care, so that they are aware of the full range of potential barriers to access to care and treatment experienced by women living with HIV. These are very likely to include violence against women at home and in health care settings, and concomitant anxiety, depression and other mental health challenges (see eg Orza et al JIAS 2015a and 2015b, Orza et al JIAS 2017, Desliets et al J Assoc Nurses AIDS Care 2020). More recent studies in the UK amongst migrant communities have clearly shown the benefits of peer mentor mother support networks run by and for women living with HIV (ie not just by external NGOs or health service providers) (eg Cameron et al 2018 https://salamanderttrust.net/wpcontent/uploads/2018/04/4M_LSHTM_BHIVA-PosterFINAL2018.pdf ; and Hay et al 2020 <https://www.tandfonline.com/doi/full/10.1080/09540121.2020.1739220>). It is also tantalising that the study was considered to be complete before there was a chance for the women concerned to

comment on whether or not the support they were receiving had improved. There is often a big gap between what women actually want, or prioritise, and what health service providers think they need. See eg 2 documents to support this: a) using the UNAIDS-commissioned ALIV[H]E framework in MENA region (p15 especially)

https://salamandertrust.net/wpcontent/uploads/2017/11/ALIVHE_in_Action_FINAL_Salamander_et_al_March2019.pdf and b) Salamander Trust, PIPE and UNYPA 2018 - see eg images on p4 https://issuu.com/salamandertrust.net/docs/20180308_4m_advocacybriefflowresfina).

Thank you for this comment. We agree that it is important to measure the effect of interventions on mothers and their children. However, this paper and study specifically and consciously : sought to assess the effect of a participatory intervention for health workers on their knowledge, attitudes and confidence. Given the study timelines we did not seek to measure the effect of the intervention on mothers' infant feeding practices. This is definitely being included in our next pieces of work, which we are seeking funding for. However, women were involved in qualitative interviews : we interviewed women who had received care from the participating clinics to explore their perceptions of care and this has been published separately: Doherty T, Horwood C, Haskins L, Magasana V, Goga A, Feucht U, et al. Breastfeeding advice for reality: Women's perspectives on primary care support in South Africa. *Maternal & Child Nutrition* 2020; 16(1):e12877 DOI: 10.1111/mcn.12877.

In addition to these overall comments, here are some more specific ones for your consideration: LANGUAGE - please be mindful of avoiding use of stigmatising and derogatory language throughout. See eg Dilmitis et al 2010 JIAS; and UNAIDS terminology guide 2015.

We have reviewed the UNAIDS 2015 document and the only terminology we found that may be a problem is MLHIV – The document states that 'People should never be referred to as an abbreviation, such as PLHIV, since this is dehumanizing. Instead, the name or identity of the group should be written out in full. Abbreviations for population groups can, however, be used in charts or graphs where brevity is required.' Thus, we have changed MLHIV to mothers living with HIV everywhere in the paper. We have also changed the words HIV infected and uninfected to HIV positive and negative, as per the UNAIDS guidance.

BEHAVIOUR CHANGE. Refs 14, 15 and 16 These are all very old behaviour change models. While the Fink has merits, the linear behaviour change model is definitely very simplistic. See also eg <https://academic.oup.com/heapro/article/34/3/616/4951539?guestAccessKey=c0cf0f69-57a3-4a35-9df9-091baa1e2ee0>

Thank you for this note. We will note this for future research. We cannot change the model on which the study was based in the write-up, and have already published the intervention paper which outlines the theories on which the intervention was based. see publication:

Christiane Horwood , Lyn Haskins, Ameena Goga, Tanya Doherty, Vaughn John, Ingunn M.S. , Engebretsen, Ute Feucht, Nigel Rollins, Max Kroon, David Sanders, Thorkild Tylleskar. An educational intervention to update health workers about HIV and infant feeding. *Maternal & Child Nutrition*. <https://doi.org/10.1111/mcn.12922>.

HEALTH STAFF - were any of the trainee health staff also living with HIV? The manuscript appears to assume not. This is a potential missed opportunity for shared insights and learning. See eg Odetoynbo et al JIAS. 2009 <https://www.ncbi.nlm.nih.gov/pmc/articles/PMC2664321/>

We did not gather this information. It may have been a missed opportunity but the primary focus of our manuscript was not to assess the relationship between health care workers' HIV status and

knowledge, It was to assess whether a participatory mentoring approach improves health care workers' knowledge of, attitudes towards and confidence with counselling on HIV and infant feeding.

TABLE 3 -

please specifically reference the source of the true information in this table for ease of access for the reader. These statements are taken from multiple sources including the Breastfeeding Counselling: A Training Course; HIV and Infant feeding counselling; a training course and the 2016 WHO Infant feeding update. These have now been referenced in the paper.

VERSION 2 – REVIEW

REVIEWER	Sara Jewett Nieuwoudt University of the Witwatersrand
REVIEW RETURNED	07-Jun-2020

GENERAL COMMENTS	I was pleased to see the revisions to this manuscript, which have strengthened the overall presentation and rigour of this work. This contributes to how we think about supporting HWs who are involved in infant feeding counselling. The Discussion, while improved, still could provide more critical engagement with the results. I've attached the revised pdf of the manuscript, where I have highlighted grammatical errors that need correction (which were numerous). For the most part, these comments focus on the manuscript's arguments and suggestions for the authors to consider prior to publication. Introduction p3, line 47: As far as I know, the guidelines state 6 months, not "about six months", but I stand to be corrected. p4, line 8: Add a question mark to the end of the question Study Design p.4, lines 22-25: Much is made of the differences between KZN and Gauteng, but then the data were pooled. What is the rationale for providing this explanation of provincial differences? p4, lines 24-5: Clarify which policies changed with the new guidelines in 2017. Policies and guidelines are not the same. Sample Size p5, line 10: Sample size calculation mentions the number of clinics, but not the number of HWs for each clinic. Add this information. Data analysis p6, line 50: The lower bands of both attitude and confidence scales are presented incorrectly. They should be 21 and 19 respectively based on the number of items.
---

	p7, line 8: Change “altitude” to “attitude” p7, line 10: Use past tense for all descriptions of analysis for consistency (see also line 16 and 19). Model 2 is discussed here, but not presented in results. As such, as it necessary to discuss Model 2 at all? Results p.10, Table 2: The statistics on how many workshops HWs attended was 249 (out of 303 eligible), but the percentage calculations treat 249 as 100%, which seems misleading. What happened to the ~18% missing data? This part of Table 2 is not discussed at all in the text, but is referred to on p.11, line 7, to argue for a dose response (data not shown) and p15, line 33 in noting there was not a significant dose response. p.13, Table 3, line 33: Percentage in the baseline comparison column has 2 decimal places p.14, Table 4 and accompanying text: As Model 2 is not presented in the article, the reference to three approaches and presentation of Model 1 and Model 3 are somewhat confusing to the reader. As with comments in data analysis, I’m wondering if this isn’t more of a distraction than necessary. If all three will be reference, maybe indicate to the reader where Model 2 is shown. p.15, Table 5 and accompanying text: Same comment about missing Model 2 in the presentation as noted for Table 4. Discussion The first paragraph does not discuss findings, but synthesizes the results. As such, it may be better placed as a concluding paragraph in the Results section. There is also a lot of similarity between this content and what is discussed in the third paragraph of the discussion. Can you synthesize? p.16, lines 28-29: Suggest responding confidence in a separate sentence, as improvements were not significant in regression analysis for Model 3. p.16, line 33: Mentions 2017 guidelines while on line 50 mentions 2016 guidelines. Correct year. The second paragraph does not discuss findings. It repeats background information on the intervention. This seems misplaced unless integrated into discussion. p.16, line 55: A deeper engagement with the literature about why HWs may have reported lower confidence for some ‘difficult’ counseling topics would strengthen this section, which does not engage with the existing literature. Some of the manuscript authors’ own work could be included here, particularly Doherty, Rollins and Horwood. Other potential articles that have engaged with HW perspectives include:  • Buskens I & Jaffe A. Demotivating infant feeding counselling encounters in southern Africa: do counsellors need more or different training? AIDS Care. 2008;20(3):337–45.
--	--

	 • Desclaux A & Alfieri C. Counseling and choosing between infant-feeding options: overall limits and local interpretations by health care providers and women living with HIV in resource-poor countries (Burkina Faso, Cambodia, Cameroon). Soc Sci Med. 2009;69:821–9. • Nieuwoudt S & Manderson L. Frontline health workers and exclusive breastfeeding guidelines in an HIV endemic South African community: a qualitative exploration of policy translation International Breastfeeding Journal (2018) 13:20 https://doi.org/10.1186/s13006-018-0164-y p.17, lines 5-6: Considering the above-mentioned work, I wonder if the authors would still hypothesize that longer durations of counselling would be able shift this confidence? An alternative hypothesis is that the HW's low confidence, at least around topics like non-adherence and high viral load, reflect more complex dynamics that are not easily addressed through counselling. p.17, line 25: The reference [30] does not refer to this study. It seems to be misplaced. p.18, line 20: The results were also not assessed in terms of how HW shifted their counselling practices (which seems like the next logical step before looking at feeding practices of mothers) p.18, line 30: "...health worker counselling practices..."
--	--

REVIEWER	Alice Welbourn Salamander Trust, UK
REVIEW RETURNED	05-Jun-2020

GENERAL COMMENTS	 1. Thank you for making other language changes, Please change the shortening of 'Prevention of Vertical Transmission' to 'PVT', not 'PMTCT'. 2. Unfortunately, the PPI statement about why you did not include women living with HIV in this study still does not align with the WHO 2017 Guideline on SRHR of women living with HIV around their meaningful involvement in any research which affects their lives. See also this article: https://www.ncbi.nlm.nih.gov/pmc/articles/PMC6803394/ about the value of involving patients for quality improvement in healthcare. Later on in the paper you also mention: "poor alignment between community views/ practice and health programmes.[29]". No matter how good a training may be, unless it produces the intended outcome amongst those who are supposed to be the end beneficiaries, it is not a complete study. This omission therefore continues to be a limitation of this study and should be identified clearly as such, in line with current standards. It should also be identified as an area for future development. 3. Thank you again for adjusting the language in places. However, It is not ideal to keep mentioning 'mothers living with HIV' - in more places than currently, they could be referred to as women living with HIV, since eg breastfeeding women are most likely to be mothers also. This is all about looking at women holistically, rather than just as reproductive machines, which feels dehumanising.
--

	4. You rightly describe the ongoing health worker crisis in limited settings - here is an opportunity also to highlight the possibility of developing a peer mentor mother programme, so that women living with HIV themselves can feel supported to share this information with each other. Recent examples from eg UNICEF's programme in Southern Africa suggests early positive results from peer mentor mother work. And I cited previously other examples. It is also in line with the WHO 2017 SRHR Guideline. This should be highlighted also for further research.
--	---

VERSION 2 – AUTHOR RESPONSE

Reviewer(s)' Comments to Author:

Reviewer: 3

Reviewer Name: Alice Welbourn

Institution and Country: Salamander Trust, UK

Please state any competing interests or state 'None declared': None declared

Please leave your comments for the authors below

1. Thank you for making other language changes, Please change the shortening of 'Prevention of Vertical Transmission' to 'PVT', not 'PMTCT'.

PVT is not a known or globally acceptable term; thus, for ease of sharing our article through MESH headings we have reverted to using the term preventing mother-to-child transmission of HIV (PMTCT) in the manuscript and have thus reverted to PMTCT. We hope this is OK. We agree that this does not and should not apportion blame on the mother. There needs to be an international dialogue about the term PMTCT and an agreement on whether to change to PVT instead of PMTCT .

2. Unfortunately, the PPI statement about why you did not include women living with HIV in this study still does not align with the WHO 2017 Guideline on SRHR of women living with HIV around their meaningful involvement in any research which affects their lives. See also this article: <https://www.ncbi.nlm.nih.gov/pmc/articles/PMC6803394/> about the value of involving patients for quality improvement in healthcare. Later on in the paper you also mention: "poor alignment between community views/ practice and health programmes.[29]". No matter how good a training may be, unless it produces the intended outcome amongst those who are supposed to be the end beneficiaries, it is not a complete study. This omission therefore continues to be a limitation of this study and should be identified clearly as such, in line with current standards. It should also be identified as an area for future development.

We completely agree that the ultimate aim is to improve feeding practices amongst mothers, and that in any work going forward we will need to collaborate with women who are HIV negative and women who are living with HIV to co-design interventions that aim to improve health workers' knowledge, attitudes, skills and practices. However, this study did not set out to do this. It was a very limited piece of work and the aims captured in the study protocol, stated that:

'This study aims to assess the feasibility and impact of a novel outreach-based mentorship approach, implemented through the primary health care system, for disseminating the 2016 updated WHO HIV and infant feeding guidelines on infant feeding attitudes and practices of front-line health workers in South Africa.'

Thus it completely focused only on health workers, and we co-designed the intervention with health workers. At this late stage we cannot change the aims of the study, nor can we not report significant results because the study did not co-design the intervention with women living with HIV – it was

unfortunately not set up to do the latter; nor was it set up to measure changes in feeding practices as the time frame was limited, the funder needed the work to be completed by a specific date. We did however include women, including those living with HIV, in a qualitative sub-study to explore their perspectives of the infant feeding counselling received at the intervention facilities. This has been published as a separate paper (Doherty T, Horwood C, Haskins L, Magasana V, Goga A, Feucht U, et al. **Breastfeeding advice for reality: Women's perspectives on primary care support in South Africa**. *Maternal & Child Nutrition* 2020; 16(1):e12877 DOI: 10.1111/mcn.12877.) We had already included this as a limitation; however we have now expanded on this and made this a separate point as it is an important limitation. We have also included this as an area for future research.

3. Thank you again for adjusting the language in places. However, It is not ideal to keep mentioning 'mothers living with HIV' - in more places than currently, they could be referred to as women living with HIV, since eg breastfeeding women are most likely to be mothers also. This is all about looking at women holistically, rather than just as reproductive machines, which feels dehumanising.

We disagree with the view that using the word 'mother/mothers' views women as reproductive machines and is thus dehumanizing, because the word mother in this context denotes someone with a child, and so it specifically indicates a particular group of women, and as a group we have respect for the position of 'mother'.. However, we have amended the term to women living with HIV.

4. You rightly describe the ongoing health worker crisis in limited settings - here is an opportunity also to highlight the possibility of developing a peer mentor mother programme, so that women living with HIV themselves can feel supported to share this information with each other. Recent examples from eg UNICEF's programme in Southern Africa suggests early positive results from peer mentor mother work. And I cited previously other examples. It is also in line with the WHO 2017 SRHR Guideline. This should be highlighted also for further research.

*This is an important point, and we agree that peer-peer mentors or feeding buddies can influence feeding practices, and we have added this point in. However the purpose of this study (and thus the purpose of this paper) was to investigate the effect of a novel outreach-based mentorship approach, implemented through the primary health care system, for disseminating the 2016 updated WHO HIV and infant feeding guidelines on infant feeding attitudes and practices of **front-line health workers** in South Africa. Thus it was outside the scope of this study to investigate the effect of peer counsellors / supporters / mentors / feeding buddies on infant feeding practices.*

Reviewer: 1

Reviewer Name: Sara Jewett Nieuwoudt

Institution and Country: University of the Witwatersrand

Please state any competing interests or state 'None declared': None declared

Please leave your comments for the authors below

I was pleased to see the revisions to this manuscript, which have strengthened the overall presentation and rigour of this work. This contributes to how we think about supporting HWs who are involved in infant feeding counselling. The Discussion, while improved, still could

provide more critical engagement with the results.

I've attached the revised pdf of the manuscript, where I have highlighted grammatical errors that need correction (which were numerous). For the most part, these comments focus on the manuscript's arguments and suggestions for the authors to consider prior to publication.

Thanks very much for these suggestions. They have helped us to clarify the intention of the study and manuscript, and have us improve the quality of the manuscript.

Introduction

p3, line 47: As far as I know, the guidelines state 6 months, not “about six months”, but I stand to be corrected.

This part has been removed. But you are correct – it should have been from 6 months

p4, line 8: Add a question mark to the end of the question

Done

Study Design

p.4, lines 22-25: Much is made of the differences between KZN and Gauteng, but then the data were pooled. What is the rationale for providing this explanation of provincial differences?

You are correct – this is not an important piece of information – it refers to historical context before 2011. but both adopted the Tshwane declaration of support for breastfeeding in 2011.

p4, lines 24-5: Clarify which policies changed with the new guidelines in 2017. Policies and guidelines are not the same.

The wording has been amended for clarity

Sample Size

p5, line 10: Sample size calculation mentions the number of clinics, but not the number of HWs for each clinic. Add this information.

Thank you for noticing this. we have added in information about sample size for health workers. Figure 3 contains more details.

Data analysis

p6, line 50: The lower bands of both attitude and confidence scales are presented incorrectly. They should be 21 and 19 respectively based on the number of items.

Thank you – this has been corrected

p7, line 8: Change “altitude” to “attitude”

Changed – thank you

p7, line 10: Use past tense for all descriptions of analysis for consistency (see also line 16 and

19). Model 2 is discussed here, but not presented in results. As such, as it necessary to discuss Model 2 at all?

The tense has been revised where appropriate, and past tense has been used when appropriate.

Results

p.10, Table 2: The statistics on how many workshops HWs attended was 249 (out of 303 eligible), but the percentage calculations treat 249 as 100%, which seems misleading. What happened to the ~18% missing data? This part of Table 2 is not discussed at all in the text, but is referred to on p.11, line 7, to argue for a dose response (data not shown) and p15, line 33 in noting there was not a significant dose response.

Apologies – we have corrected the results on number of workshops in Table 2. Table 2 is descriptive, and the role of dose is referred to later on in the results. This table was added in so that the reader understands a bit more about the workshops.

p.13, Table 3, line 33: Percentage in the baseline comparison column has 2 decimal places

Thank you – this has been corrected

p.14, Table 4 and accompanying text: As Model 2 is not presented in the article, the reference to three approaches and presentation of Model 1 and Model 3 are somewhat confusing to the reader. As with comments in data analysis, I'm wondering if this isn't more of a distraction than necessary. If all three will be reference, maybe indicate to the reader where Model 2 is shown.

We were keen to present all three methods in the data analysis section, to the reader to illustrate that the findings are robust, regardless of the analytical methods: However, we have amended the wording to include three methods but present 2 models because methods 1 and 2 yielded very similar responses. we have amended the wording as follows:

- *Method 1 used the post-treatment measurements as the outcome variable, but adjusted for the pre-treatment values;*
- *Method 2 analysed the change score as an outcome variable adjusting for pre-treatment values;*
- *Method 3 analysed all the pre-and post-measurements as the outcome variable, and used time (coded : 1 at follow-up and 0 at baseline) as a covariate with an interaction term for time and treatment, in addition to an adjustment for pre-treatment values).*

p.15, Table 5 and accompanying text: Same comment about missing Model 2 in the presentation as noted for Table 4.

We have amended the wording – hope it's clearer now.

Discussion

The first paragraph does not discuss findings, but synthesizes the results. As such, it may be better placed as a concluding paragraph in the Results section. There is also a lot of similarity between this content and what is discussed in the third paragraph of the discussion. Can you synthesize?

Thanks for this comments – the paragraphs have been synthesised

p.16, lines 28-29: Suggest responding confidence in a separate sentence, as improvements were not significant in regression analysis for Model 3.

Done

p.16, line 33: Mentions 2017 guidelines while on line 50 mentions 2016 guidelines. Correct year.

We have clarified that the WHO guidelines were updated in 2016 and the SA policy was revised in June 2017. The reference to the two guidelines have now been clarified in the paper.

The second paragraph does not discuss findings. It repeats background information on the intervention. This seems misplaced unless integrated into discussion.

We have now included description of the features of the mentoring approach into the methods section.

p.16, line 55: A deeper engagement with the literature about why HWs may have reported lower confidence for some 'difficult' counseling topics would strengthen this section, which does not engage with the existing literature. Some of the manuscript authors' own work could be included here, particularly Doherty, Rollins and Horwood. Other potential articles that have engaged with HW perspectives include:

- Buskens I & Jaffe A. Demotivating infant feeding counselling encounters in southern Africa: do counsellors need more or different training? *AIDS Care*. 2008;20(3):337–45.
- Desclaux A & Alfieri C. Counseling and choosing between infant-feeding options: overall limits and local interpretations by health care providers and women living with HIV in resource-poor countries (Burkina Faso, Cambodia, Cameroon). *Soc Sci Med*. 2009;69:821–9.
- Nieuwoudt S & Manderson L. Frontline health workers and exclusive breastfeeding guidelines in an HIV endemic South African community: a qualitative exploration of policy translation *International Breastfeeding Journal* (2018) 13:20 <https://doi.org/10.1186/s13006-018-0164-y>

Thanks for this comment – we have engaged with this issue a bit more and added in additional references. We have also amended the introduction to clarify the aims of the study.

p.17, lines 5-6: Considering the above-mentioned work, I wonder if the authors would still hypothesize that longer durations of counselling would be able shift this confidence? An alternative hypothesis is that the HW's low confidence, at least around topics like non-adherence and high viral load, reflect more complex dynamics that are not easily addressed through counselling.

This is quite true – we have added this in. Thanks so much for raising this. We have added this.

p.17, line 25: The reference [30] does not refer to this study. It seems to be misplaced.

*The reference is Schwerdt P, Mophet, Hall H. A scoping review of mentorship of health personnel to improve the quality of health care in low and middle-income countries. *Globalisation and Health* 2017;13(77):doi 10.1186/s12992-017-0301-1*

We have gone back to review the reference and don't understand why the reviewer believes that the reference is misplaced. This is a scoping review and it raises important points about mentorship, which are referred to in the discussion.

p.18, line 20: The results were also not assessed in terms of how HW shifted their counselling practices (which seems like the next logical step before looking at feeding practices of mothers)

This limitation has been added

p.18, line 30: "...health worker counselling practices..." *This has been amended*

VERSION 3 – REVIEW

REVIEWER	Sara Jewett Nieuwoudt University of the Witwatersrand
REVIEW RETURNED	03-Jul-2020

GENERAL COMMENTS	It was a pleasure to read the latest version of this manuscript. The authors have addressed all of my substantive issues and I am particularly impressed by the revisions to the Discussion. I picked up a few minor editing issues to be addressed prior to final submission, but I feel this is now ready to be published. These are highlighted in the attachment and noted below. ABSTRACT p2, line 26: remove extra) p2, line 30: add 'the' before attitude score p2, lines 42-43: consider rewording phrase 'routine facilities' to 'routine facility services' or something similar INTRODUCTION p3, line 30: remove . after references p3, line 35: add references for 2017 IYCF policy p3, line 45: correct spelling of 'strategies' p4, lines 10-11: check spacing between . and references SAMPLE SIZE p5, line 10: CHW already introduced. Do not reintroduce acronym. DESCRIPTION OF INTERVENTION p5, line 34: Theory of Planned Behaviour (proper noun) PUBLIC INVOLVEMENT p6, line 14: Remove extra . DATA COLLECTION p6, line 23: Add 'the' before confidence p6, lines 25 and 26: Check highlighted punctuation/spacing DATA ANALYSIS p7, lines 16-18: Check highlighted punctuation/spacing RESULTS
---

	p7, line 46: Ensure both percentages have same specificity (1 or 0 decimals) and also correct this in Figure 3 p7, line 51: Remove extra : TABLE 2 p9, line 48: Were the HWs who did not attend any workshops excluded from the dose response calculations reported later in the results? It is unclear how they were treated for analysis given lack of exposure to the workshops. INTERVENTION ON ATTITUDES p13, line 47: Only method 1 is mentioned here whereas earlier method 2 is also reported. Try to be consistent about whether methods 1&2 are being reported together throughout or explain why method 2 is not mentioned sometimes. [NB: in some places Method is in caps and in others not; be consistent] p13, line 52: Dose response measurement options are not mutually exclusive with 1 included in 0-1 as well as 1-2. Correct this description based on how this was actually measured. CONFIDENCE RESULTS p14, lines 38-55: Check highlighted punctuation/spacing DISCUSSION p.15, line 53: Check highlighted punctuation p.16, line 8: Meaning of 'pervasive broadcast' is unclear. Consider rephrasing. p.16, lines 33-36: Check highlighted punctuation/spacing p.17, lines 28 & 36: Check highlighted punctuation/spacing REFERENCES Review highlighted errors throughout Correct spelling of #10 author Martin-Wiesner Add Manderson L to #22 NB: #40 is a repeat of reference #11
--	---

REVIEWER	Alice Welbourn Salamander Trust, UK
REVIEW RETURNED	19-Jul-2020

GENERAL COMMENTS	In response to your comments dated 25th June: 1) It does not feel OK for you to revert to 'PMTCT". If you say Prevention of Vertical Transmission, if you don't want to contribute to promoting its use by adding (PVT) then please state that this is "also know as PMTCT" - The WHO Guideline clearly states: "Comprehensive prevention of "vertical transmission" can be used instead of saying "mother-to-child transmission" (or MTCT), to reduce possible blame that women living with HIV may experience. This is central to creating an environment that promotes SRHR (13)" You have it in your power to choose whether or not to help to promote this here by using 'PVT' as a way of introducing it to a wider audience. There has been a very clear international dialogue already, which is why it states this in the Guideline. See also: https://onlinelibrary.wiley.com/doi/full/10.7448/IAS.15.4.17990 and https://salamandertrust.net/wp-content/uploads/2016/04/4M_July2020_Advocacy_Brief_2page_summary_final.pdf and https://blogs.bmj.com/bmj/srh/2020/06/25/include-perspectives-hiv/
---

	2) Thank you for expanding on this limitation. The challenges experienced with developing the confidence of health workers to support women with the issues described at the bottom of page 14 would be reduced if the programme had consulted with women living with HIV themselves about how to address these issues with their peers. Indeed the whole knowledge dissemination issue would be more effective if the women themselves were actively engaged in discussions around their own healthcare and that of their babies. 3) Thank you for referring to women rather than mothers here. 4) Thank you for adding this point in. However, in your tracked change on p17, it looks as if there is an either/or between mentoring for HW or peer mentoring by women living with HIV. Whilst I appreciate you were only exploring the former, it would surely be preferable to be promoting both here, given point 2 above especially.
--	---

VERSION 3 – AUTHOR RESPONSE

Reviewer: 1

Reviewer Name: Sara Jewett Nieuwoudt

Institution and Country: University of the Witwatersrand

Please state any competing interests or state 'None declared': None Declared

Please leave your comments for the authors below

It was a pleasure to read the latest version of this manuscript. The authors have addressed all of my substantive issues and I am particularly impressed by the revisions to the Discussion.

I picked up a few minor editing issues to be addressed prior to final submission, but I feel this is now ready to be published. These are highlighted in the attachment and noted below.

ABSTRACT

p2, line 26: remove extra) *Done*

p2, line 30: add 'the' before attitude score *Done*

p2, lines 42-43: consider rewording phrase 'routine facilities' to 'routine facility services' or something similar *Done*

INTRODUCTION

p3, line 30: remove . after references *Done*

p3, line 35: add references for 2017 IYCF policy – the infant and young child feeding policy has not been formally updated. The update was released on a circular, so there is no reference for it.

p3, line 45: correct spelling of 'strategies' *Done*

p4, lines 10-11: check spacing between . and references *Done*

SAMPLE SIZE

p5, line 10: CHW already introduced. Do not reintroduce acronym. *Done*

DESCRIPTION OF INTERVENTION

p5, line 34: Theory of Planned Behaviour (proper noun) *Done*

PUBLIC INVOLVEMENT

p6, line 14: Remove extra . *Done*

DATA COLLECTION

p6, line 23: Add 'the' before confidence *Done*

p6, lines 25 and 26: Check highlighted punctuation/spacing *Done*

DATA ANALYSIS

p7, lines 16-18: Check highlighted punctuation/spacing *Done*

RESULTS

p7, line 46: Ensure both percentages have same specificity (1 or 0 decimals) and also correct this in Figure 3 *Done*

p7, line 51: Remove extra : *Done*

TABLE 2

p9, line 48: Were the HWs who did not attend any workshops excluded from the dose response calculations reported later in the results? It is unclear how they were treated for analysis given lack of exposure to the workshops.

The analyses of treatment effect comparison presented in Tables 4 and 5 were based on Intention to treat (ITT) analysis (as opposed to a per-protocol (PP) analysis). Thus, HWs who did not attend any workshop were not excluded from the treatment group in estimating the mentorship program effect. We have removed results concerning dose response effect from the narrative pertaining to Table 4 and 5, and instead indicated these at the end of the results section. The dose response analysis included attendance at 0, 1 or 2, or 3 workshops.

INTERVENTION ON ATTITUDES

p13, line 47: Only method 1 is mentioned here whereas earlier method 2 is also reported. Try to be consistent about whether methods 1&2 are being reported together throughout or explain why method 2 is not mentioned sometimes. [NB: in some places Method is in caps and in others not; be consistent]

changed and clarified

p13, line 52: Dose response measurement options are not mutually exclusive with 1 included in 0-1 as well as 1-2. Correct this description based on how this was actually measured.

This has now been changed and clarified

CONFIDENCE RESULTS

p14, lines 38-55: Check highlighted punctuation/spacing *Done*

DISCUSSION

p.15, line 53: Check highlighted punctuation *Done*

p.16, line 8: Meaning of 'pervasive broadcast' is unclear. Consider rephrasing. *Apologies – this has been re-phrased*

p.16, lines 33-36: Check highlighted punctuation/spacing *Done*

p.17, lines 28 & 36: Check highlighted punctuation/spacing *Done*

REFERENCES

Review highlighted errors throughout

Correct spelling of #10 author Martin-Wiesner - changed

Add Manderson L to #22 - changed

NB: #40 is a repeat of reference #11

Thanks so much – references have been corrected

Reviewer: 3

Reviewer Name: Alice Welbourn

Institution and Country: Salamander Trust, UK

Please state any competing interests or state 'None declared': None declared

Please leave your comments for the authors below

In response to your comments dated 25th June:

1) It does not feel OK for you to revert to 'PMTCT'. If you say Prevention of Vertical Transmission, if you don't want to contribute to promoting its use by adding (PVT) then please state that this is "also known as PMTCT" - The WHO Guideline clearly states: "Comprehensive prevention of "vertical transmission" can be used instead of saying "mother-to-child transmission" (or MTCT), to reduce possible blame that women living with HIV may experience. This is central to creating an environment that promotes SRHR (13)" You have it in your power to choose whether or not to help to promote this here by using 'PVT' as a way of introducing it to a wider audience. There has been a very clear international dialogue already, which is why it states this in the Guideline. See also:

<https://onlinelibrary.wiley.com/doi/full/10.7448/IAS.15.4.17990> and https://salamandertrust.net/wp-content/uploads/2016/04/4M_July2020_Advocacy_Brief_2page_summary_final.pdf and <https://blogs.bmj.com/bmj/rh/2020/06/25/include-perspectives-hiv/>

Thank you for insisting on this change – we need to get used to the new terminology - we have added in PVT. However we have left PMTCT where it refers to the names of documents as we cannot change the names of documents.

2) Thank you for expanding on this limitation. The challenges experienced with developing the confidence of health workers to support women with the issues described at the bottom of page 14 would be reduced if the programme had consulted with women living with HIV themselves about how to address these issues with their peers. Indeed the whole knowledge dissemination issue would be more effective if the women themselves were actively engaged in discussions around their own healthcare and that of their babies. Thank you for raising this point, we agree with you and will bear that in mind when planning future work. We have added in this point in the discussion

3) Thank you for referring to women rather than mothers here. Pleasure

4) Thank you for adding this point in. However, in your tracked change on p17, it looks as if there is an either/or between mentoring for HW or peer mentoring by women living with HIV. Whilst I appreciate you were only exploring the former, it would surely be preferable to be promoting both here, given point 2 above especially. This point has now been added to the discussion.